# Debiased Front-Door Learners for Heterogeneous Effects

**Yonghan Jung**
School of Information Sciences
University of Illinois Urbana-Champaign
yonghan@illinois.edu

## Abstract

In observational settings where treatment and outcome share unmeasured confounders but an observed mediator remains unconfounded, the front-door (FD) adjustment identifies causal effects through the mediator. We study the *heterogeneous treatment effect* (HTE) under FD identification and introduce two debiased learners: *FD-DR-Learner* and *FD-R-Learner*. Both attain fast, quasi-oracle rates (i.e., performance comparable to an oracle that knows the nuisances) even when nuisance functions converge as slowly as $n^{-1/4}$. We provide error analyses establishing debiasedness and demonstrate robust empirical performance in synthetic studies and a real-world case study of primary seat-belt laws using Fatality Analysis Reporting System (FARS) dataset. Together, these results indicate that the proposed learners deliver reliable and sample-efficient HTE estimates in FD scenarios. The implementation is available at https://github.com/yonghanjung/FD-CATE.

**Keywords:** Front-door adjustment, Heterogeneous treatment effects, Debiased learning, Quasi-oracle rates, DR-Learner, R-Learner

## 1 Introduction

Estimating causal effects from observational data is central to disciplines such as public policy. A crucial challenge is unmeasured confounding, where the treatment a unit receives is influenced by unobserved variables that also affect the outcome. The *front-door* (FD) criterion (Pearl, 1995) addresses this by using an observed mediator that transmits the treatment's influence to the outcome and is plausibly explained by observed covariates.

A concrete instance (which we also use in our empirical study in § 5.2) is the effect of adopting a primary seat-belt law ($X$) on occupant fatality ($Y$), with observed seat-belt use ($Z$) as the mediator, and observed covariate $C$. This scenario is depicted in Fig. 1a. Because of unmeasured confounding between $X$ and $Y$, the naive contrast $\mathbb{E}[Y \mid X = 1] - \mathbb{E}[Y \mid X = 0]$ is a biased estimate of the causal effect. When belt use responds strongly to law adoption and the observed covariates plausibly explain belt use, the setting is consistent with an FD structure, enabling identification and estimation of the causal effect via FD adjustment (Pearl, 1995).

Although robust FD estimation has advanced recently (refer § 1.1), most methods target population averages (the *average treatment effect*). In practice, platforms require *personalized* effects—i.e., the conditional front-door effect $\tau(C)$—to support decision-making at the user or context level. We address this gap by adapting heterogeneous treatment effect estimators developed for the standard ignorability (Rubin, 1974) (or back-door adjustment (Pearl, 1995)), such as the DR-Learner (Kennedy, 2023) and the R-learner (Nie & Wager, 2021), to the front-door setting. Concretely,

1. **FD-DR-Learner.** We construct a pseudo-outcome whose conditional mean equals $\tau(C)$. Regressing this pseudo-outcome on $C$ yields a debiased estimator that achieves quasi-

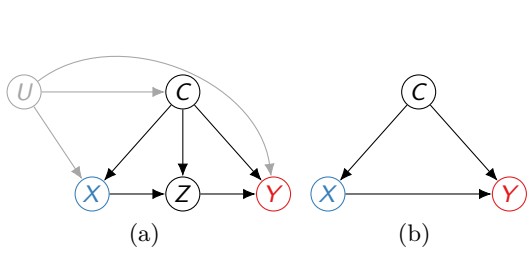

| | FD | HTE | Debiasedness |
|---|---|---|---|
| Nie & Wager (2021)
Kennedy (2023) | ✗ | ✓ | ✓ |
| Xu & Gretton (2022)
Xu et al. (2024) | ✓ | ✗ | ✗ |
| Fulcher et al. (2019)
Guo et al. (2023)
Wen et al. (2024)
Jung et al. (2024) | ✓ | ✗ | ✓ |
| Chen et al. (2025) | ✓ | ✓ | ✗ |
| **Ours** | ✓ | ✓ | ✓ |

(c)

Figure 1: **(a)** Causal diagram illustrating the front-door (FD) structure; **(b)** back-door (BD) structure; **(c)** comparison table indicating whether methods estimate FD effects, heterogeneous treatment effects (HTE), and are debiased. Our proposed learners satisfy all three.

    oracle rates (the rate achievable with true nuisances) whenever the nuisances converge at $n^{-1/4}$-rates.

2. **FD-R-Learner.** We develop a three-stage procedure: (1) estimate how $X$ changes $Z$ across $C$; (2) estimate how changes in $Z$ shift $Y$ given $(X, C)$; (3) compose these estimates to obtain $\tau(C)$. This estimator achieves quasi-oracle rates whenever nuisances converge at $n^{-1/4}$-rates.

3. **Theory and Practice.** We provide error analyses of the proposed estimators, showing debiasedness under slow nuisance convergence. We demonstrate our findings with synthetic and real-world data analysis.

This paper is organized as follows. §3-4 presents the FD-DR and FD-R learners. §5 reports simulations and a case study. All proofs are deferred to Section B.

## 1.1 Related works

For back-door adjustment (g-formula), a large literature establishes debiased and sample-efficient estimation. Classical approaches include augmented inverse probability weighting (Robins et al., 1994; Bang & Robins, 2005), as well as targeted maximum likelihood estimation (TMLE) (Van Der Laan & Rubin, 2006; van der Laan & Gruber, 2012). More recently, estimation frameworks leveraging machine learning for heterogeneous treatment effect have produced flexible estimators with finite-sample guarantees, notably the DR-Learner (Kennedy, 2023), the R-learner (Nie & Wager, 2021), the EP-learner (van der Laan et al., 2024), orthogonal statistical learning (Foster & Syrgkanis, 2023), and double/debiased machine learning (Chernozhukov et al., 2018).

Beyond back-door adjustment, research on developing debiased estimators for the front-door (FD) model has grown steadily. Fulcher et al. (2019) develop a doubly robust estimator for the FD average treatment effect estimator. Guo et al. (2023) propose one–step and TMLE estimators for FD adjustment. Jung et al. (2024) introduce a unified covariate-adjustment formulation that enables robust and sample-efficient FD estimation. On a different thread, modern deep learning-based FD estimators have been developed for scalable estimation (Xu & Gretton, 2022; Xu et al., 2024), but these methods lack the debiasedness property.

Beyond average effects, work on *heterogeneous* or *conditional* FD has also emerged. Chen et al. (2025) study conditional FD and introduce *LobsterNet*, a multi-task neural estimator for the conditional FD effect. However, their estimator lacks debiasedness. In this work, we propose debiased learners for heterogeneous FD treatment effects, which allow any machine learning method to be used off the shelf while guaranteeing fast convergence to the target even when nuisance parameters converge slowly. A summary comparing existing works with ours is in Table 1c.

## 2 PROBLEM SETUP

We observe i.i.d. samples of $V = (C, X, Z, Y)$, where $C$ is observed covariates, $X \in \{0, 1\}$ a binary treatment, $Z \in \{0, 1\}$ a binary mediator, and $Y$ is the outcome. Let $U$ represent an unobserved variable that jointly influences $(C, X, Y)$. We define the nuisance functions

$$m(zxc) \triangleq \mathbb{E}[Y \mid Z = z, X = x, C = c], \tag{1}$$

$$e(x \mid c) \triangleq \Pr(X = x \mid C = c) \quad \text{with } e_1(C) \triangleq e(1 \mid C), \text{ and } e_0(C) \triangleq e(0 \mid C). \tag{2}$$

$$q(z \mid xc) \triangleq \Pr(Z = z \mid X = x, C = c) \tag{3}$$

We assume positivity: $0 < e(x \mid c), q(z \mid xc) < 1$ for $x, z \in \{0, 1\}$ and almost every $c$. We also assume $\mathrm{Var}(Y) < \infty$.

The data-generating process is depicted in the causal diagram $\mathcal{G}$ in Fig. 1a. Specifically, the structure satisfies the conditional front-door criterion (Pearl, 1995; Fulcher et al., 2019):

1. Every directed path from $X$ to $Y$ is mediated by $Z$ in $\mathcal{G}$ (no unmediated direct effect).
2. Every spurious path between $X$ and $Z$ is blocked (d-separated) by $C$ in $\mathcal{G}$.
3. Every spurious path between $Z$ and $Y$ is blocked (d-separated) by $(X, C)$ in $\mathcal{G}$.

Let $\tau_{\bar{x}}(C) \triangleq \mathbb{E}[Y \mid \mathrm{do}(X = \bar{x}), C]$, where $\mathrm{do}(X = \bar{x})$ denotes an intervention that fixes $X$ to $\bar{x}$. Under the graph in Fig. 1a, the conditional treatment effect $\tau(C)$ is identified by

$$\tau(C) \triangleq \tau_1(C) - \tau_0(C) = \sum_{z,x} \{q(z \mid 1C) - q(z \mid 0C)\} e_x(C) m(zxC), \tag{4}$$

### 2.1 PRELIMINARIES: R-LEARNER FOR BACK-DOOR ADJUSTMENT (BD-R-LEARNER)

In this subsection, we review the standard R-learner of (Nie & Wager, 2021), a key building block for developing our FD-R-Learner. We refer to this standard R-learner as *BD-R-Learner* (shortly, BDR), since it is developed under the back-door (BD) criterion (Pearl, 1995), as depicted in Fig. 1b.

To review the BD-R-Learner, define the nuisance functions $\eta \triangleq \{e_X, m_Y\}$, where

$$e_X(C) \triangleq e(1 \mid C) \quad \text{and} \quad m_Y(C) \triangleq \mathbb{E}[Y \mid C]. \tag{5}$$

Let $\tau_{\bar{x}}^{\mathrm{BD}}(C) \triangleq \mathbb{E}[Y \mid \mathrm{do}(X = \bar{x}), C]$ in the BD setting. It is identified as

$$\tau^{\mathrm{BD}}(C) \triangleq \tau_1^{\mathrm{BD}}(C) - \tau_0^{\mathrm{BD}}(C) = \mathbb{E}[Y \mid X = 1, C] - \mathbb{E}[Y \mid X = 0, C]. \tag{6}$$

In the BD graph in Fig. 1b, the data generating process can be expressed as the following partial linear model (Robinson, 1988):

$$X = e_X(C) + \epsilon_X, \quad \mathbb{E}[\epsilon_X \mid C] = 0; \quad \text{and} \tag{7}$$

$$Y = a(C) + Xb(C) + \epsilon_Y, \quad \mathbb{E}[\epsilon_Y \mid XC] = 0, \tag{8}$$

where $a(C) = \tau_0^{\mathrm{BD}}(C)$ and $b(C) = \tau^{\mathrm{BD}}(C)$.

The BD-R-learner's loss function, equipped with an arbitrary nuisance $\widetilde{\eta} = \{\widetilde{e}_X, \widetilde{m}_Y\}$ is defined as:

$$\ell_\lambda^{\mathrm{BDR}}((Y, X, C), \widetilde{\eta}, \tau) \triangleq (Y - \widetilde{m}_Y(C) - \{X - \widetilde{e}_X(C)\} \tau(C))^2. \tag{9}$$

Population and empirical risk functions for samples $\mathcal{D}(Y, X, C) \triangleq \{(Y_i, X_i, C_i)\}_{i=1}^{|\mathcal{D}|}$ are give by:

$$L_\lambda^{\mathrm{BDR}}(\tau, \widetilde{\eta}) \triangleq \mathbb{E}_{Y,X,C \sim P}[\ell_\lambda^{\mathrm{BDR}}((Y, X, C), \widetilde{\eta}, \tau)] + \lambda \mathcal{J}(\tau) \tag{10}$$

$$\widehat{L}_{\lambda, \mathcal{D}(Y,X,C)}^{\mathrm{BDR}}(\tau, \widetilde{\eta}) \triangleq \frac{1}{|\mathcal{D}|} \sum_{i:V_i \in \mathcal{D}} \ell_\lambda^{\mathrm{BDR}}((Y_i, X_i, C_i), \widetilde{\eta}, \tau) + \lambda \mathcal{J}(\tau), \tag{11}$$

where $\lambda$ is a hyperparameter and $\mathcal{J}$ is a regularizer. BD-R-Learner is defined as follows:

**Definition 1** (**BD-R-Learner** (Nie & Wager, 2021; Foster & Syrgkanis, 2023)).
The BD-R-Learner $\widehat{\tau}^{\mathrm{BDR}}$ is learned from the following procedure:

1. Split the i.i.d. dataset $\mathcal{D}(Y, X, C) \triangleq \{(Y_i, X_i, C_i)\}_{i=1}^{2n}$ into $\mathcal{D}_1$ and $\mathcal{D}_2$.
2. Learn $\widehat{\eta} \triangleq \{\widehat{e}_X, \widehat{m}_Y\}$ using $\mathcal{D}_1$.
3. Find $\widehat{\tau}^{\mathrm{BDR}} \in \arg\min_{\tau \in \mathcal{T}} \widehat{L}_{\lambda, \mathcal{D}_2(YXC)}^{BDR}(\tau, \widehat{\eta})$.

One may repeat steps (2–3) with the partitions swapped and average the two estimates. The debiasedness property of the BD-R-learner is captured by the following result:

**Proposition 1** (**Error Analysis of BD-R-Learner** (Nie & Wager, 2021; Foster & Syrgkanis, 2023)). Let $\| \cdot \|_p$ denote the $L_p(P)$ norm. Let $\widehat{\tau}^{\mathrm{BDR}}$ denote the BD-R-Learner estimate from Def. 1. Let $a \lesssim b$ denote $a \leq b$ up to a constant factor. Suppose there exists a function $\mathcal{R}_{\mathcal{T}}^{\mathrm{BDR}}(\epsilon, \widetilde{\tau}, \widetilde{\eta})$ such that, with probability $1 - \epsilon$,

$$|L_\lambda^{\mathrm{BDR}}(\widetilde{\tau}, \widetilde{\eta}) - \min_{\tau \in \mathcal{T}} L_\lambda^{\mathrm{BDR}}(\tau, \widetilde{\eta})| \leq \mathcal{R}_{\mathcal{T}}^{\mathrm{BDR}}(\epsilon, \widetilde{\tau}, \widetilde{\eta}). \tag{12}$$

Then, with probability $1 - \epsilon$,

$$\|\widehat{\tau}^{\mathrm{BDR}} - \tau^{\mathrm{BDR}}\|_2^2 \lesssim \mathcal{R}_{\mathcal{T}}^{\mathrm{BDR}}(\epsilon, \widehat{\tau}^{\mathrm{BDR}}, \widehat{\eta}) + \|\widehat{e}_X - e_X\|_4^4 + \|\widehat{m}_Y - m_Y\|_4^2 \|\widehat{e}_X - e_X\|_4^2. \tag{13}$$

This result exhibits the property of debiasedness: $\widehat{\tau}^{\mathrm{BDR}}$ converges at a quasi-oracle rate $\mathcal{R}^{\mathrm{BDR}}$ even when $\widehat{e}_X$ converges only at $n^{-1/4}$.

## 3 FD-DR-LEARNER

We first introduce the FD-DR-Learner, a learner for the conditional FD effect $\tau(C)$ that exhibits a *double robustness* property. In addition to the nuisance functions $m(ZXC)$, $e(X \mid C)$ and $q(Z \mid XC)$ in Eq. (1), (2), (3), define the following weights:

$$\xi_{\bar{x}}(ZXC) \triangleq \frac{q(Z \mid \bar{x}C)}{q(Z \mid XC)}, \quad \pi_{\bar{x}}(XC) \triangleq \frac{\mathbb{I}(X = \bar{x})}{e(X \mid C)}. \tag{14}$$

We also define the following functionals of nuisances $(m, e, q)$:

$$r_{me}(ZC) \triangleq \sum_{x \in \{0,1\}} m(ZxC)e(x \mid C), \tag{15}$$

$$\nu_{meq}(XC) \triangleq \sum_{z \in \{0,1\}} r_{me}(zC)q(z \mid XC), \tag{16}$$

$$s_{mq_{\bar{x}}}(XC) \triangleq \sum_{z \in \{0,1\}} m(zXC)q(z \mid \bar{x}C). \tag{17}$$

These induced functionals are sufficient to express the front–door estimand.

**Lemma 1** (**Expressiveness of nuisances**). For $\bar{x} \in \{0,1\}$

$$\mathbb{E}[\xi_{\bar{x}}(ZXC)Y] = \mathbb{E}[\pi_{\bar{x}}(XC)r_{me}(ZC)] = \mathbb{E}[s_{mq_{\bar{x}}}(XC)] = \mathbb{E}[\tau_{\bar{x}}(C)]. \tag{18}$$

Based on these nuisances, we define the pseudo-outcome for the FD-DR-Learner.

**Definition 2** (**Front-door Pseudo-Outcome** (FDPO)). For $\bar{x} \in \{0,1\}$, let $\eta = \{m, e, q\}$. The *front-door pseudo-outcome* (FDPO) at $\bar{x}$ is denoted as $\varphi_{\bar{x}}(V; \eta)$: d

$$\xi_{\bar{x}}(ZXC)\{Y - m(ZXC)\} + \pi_{\bar{x}}(XC)\{r_{me}(ZC) - \nu_{meq}(XC)\} + s_{mq_{\bar{x}}}(XC). \tag{19}$$

FDPO enjoys the following identification and robustness properties.

---

**Lemma 2** (**FDPO Property**). For $\bar{x} \in \{0, 1\}$, let $\eta = \{m, e, q\}$,

1. **Consistency**: $\tau_{\bar{x}}(C) = \mathbb{E}[\varphi_{\bar{x}}(V; \eta) \mid C]$; and

2. **Double Robustness**: Let $\hat{\eta}$ be an estimate of $\eta$. Let $e_x \triangleq e(X \mid C)$ and $q_{z\bar{x}} \triangleq q(z \mid \bar{x}C)$.

$$\mathbb{E}[\varphi_{\bar{x}}(V; \hat{\eta}) - \varphi_{\bar{x}}(V; \eta)] \tag{20}$$

$$= \mathbb{E}[\{\widehat{m} - m\}\{\xi_{\bar{x}} - \widehat{\xi}_{\bar{x}}\}] + \mathbb{E}[\{\widehat{\pi}_{\bar{x}} - \pi_{\bar{x}}\}\{\nu_{\widehat{mqe}} - \nu_{\widetilde{mqe}}\}] \tag{21}$$

$$+ \sum_{zx} \mathbb{E}[\widehat{m}(zxC)\{q(z \mid \bar{x}C) - \widehat{q}(z \mid \bar{x}C)\}\{\widehat{e}(x \mid C) - e(x \mid C)\}]. \tag{22}$$

---

We learn $\tau(C)$ by regressing the *difference* between the FDPO at $\bar{x} = 1$ and $\bar{x} = 0$ on $C$. Specifically, define the population and empirical risk functions for the FD-DR-Learner as

$$\mathcal{L}_\lambda^{\mathrm{DR}}(\tilde{\tau}, \tilde{\eta}) \triangleq \mathbb{E}[\{\varphi_1(V; \tilde{\eta}_1) - \varphi_0(V; \tilde{\eta}_0) - \tilde{\tau}(C)\}^2] + \lambda \mathcal{J}(\tilde{\tau}), \tag{23}$$

$$\widehat{\mathcal{L}}_{\lambda, \mathcal{D}}^{\mathrm{DR}}(\tilde{\tau}, \tilde{\eta}) \triangleq \frac{1}{|\mathcal{D}|} \sum_{i: V_i \in \mathcal{D}} \left(\{\varphi_1(V_i; \tilde{\eta}_1) - \varphi_0(V_i; \tilde{\eta}_0) - \tilde{\tau}(C_i)\}^2 + \lambda \mathcal{J}(\tilde{\tau})\right), \tag{24}$$

where $\tilde{\eta}$ denotes any nuisance set, $\tilde{\tau}$ is any candidate function, $\varphi_{\bar{x}}$ is the FDPO for $\bar{x} \in \{0, 1\}$, $\lambda \geq 0$ is a regularization parameter, and $\mathcal{J}$ penalizes model complexity.

Using the risk functions in Eqs. (23) and (24), we define the FD-DR-Learner as follows:

---

**Definition 3** (**FD-DR-Learner**). Let $\mathcal{D}_1, \mathcal{D}_2$ be two disjoint (independent) splits of size $n$ of $V_i = (C_i, X_i, Z_i, Y_i)$. Let $\mathcal{T}$ be the model class for $\tau$, and let $\lambda_n$ be a regularization level.

1. Fit $\hat{\eta} \triangleq \{\hat{m}, \hat{q}, \hat{e}\}$ on $\mathcal{D}_1$.

2. Compute $\hat{\tau}_{\mathrm{DR}} \in \arg\min_{\tilde{\tau} \in \mathcal{T}} \widehat{\mathcal{L}}_{\lambda_n, \mathcal{D}_2}^{\mathrm{DR}}(\tilde{\tau}, \hat{\eta})$ using $\mathcal{D}_2$.

3. (Optional) Swap the roles of $\mathcal{D}_1$ and $\mathcal{D}_2$ (cross-fitting) to obtain a second estimate $\hat{\tau}'$, and return the average $\hat{\tau}_{\mathrm{DR}} = (\hat{\tau} + \hat{\tau}')/2$.

---

Finally, the following result formalizes the debiasedness of the learner:

---

**Theorem 1** (**Error-Analysis of FD-DR-Learner**). Let $\mathcal{T}$ be the model class for any $\tilde{\tau}$. Suppose there exists a quasi-oracle rate function $\mathcal{R}^{\mathrm{DR}}\mathcal{T}(\epsilon, \tilde{\tau}, \tilde{\eta})$ such that, with probability at least $1 - \epsilon$,

$$|L_\lambda^{\mathrm{DR}}(\tilde{\tau}, \tilde{\eta}) - \min_{\tilde{\tau} \in \mathcal{T}} L_\lambda^{\mathrm{DR}}(\tilde{\tau}, \tilde{\eta})| \leq \mathcal{R}_{\mathcal{T}}^{\mathrm{DR}}(\epsilon, \tilde{\tau}, \tilde{\eta}) \quad \text{for all } \tilde{\tau} \in \mathcal{T}. \tag{25}$$

Let $\hat{\tau}_{\mathrm{DR}}$ denote the FD-DR-Learner. With probability at least $1 - \epsilon$,

$$\|\hat{\tau}_{\mathrm{DR}} - \tau\|_2^2 \lesssim \mathcal{R}_{\mathcal{T}}^{\mathrm{DR}}(\epsilon, \hat{\tau}_{\mathrm{DR}}, \hat{\eta}) + \sum_{\bar{x} \in \{0,1\}} \|\hat{m} - m\|_2^2 \|\hat{\xi}_{\bar{x}} - \xi_{\bar{x}}\|_2^2 \tag{26}$$

$$+ \sum_{\bar{x} \in \{0,1\}} \|\nu_{\hat{m}\hat{e}\hat{q}} - \nu_{\hat{m}\hat{e}q}\|_2^2 \|\hat{\pi}_{\bar{x}} - \pi_{\bar{x}}\|_2^2 \tag{27}$$

$$+ \sum_{\bar{x}, x, z \in \{0,1\}^3} \|\hat{e}_x - e_x\|_2^2 \|q_{z\bar{x}} - \hat{q}_{z\bar{x}}\|_2^2. \tag{28}$$

---

In words, the error of $\hat{\tau}_{\mathrm{DR}}$ is controlled by the quasi-oracle rate and second-order products of nuisance errors, reflecting the method's debiasedness. For example, if all nuisances converge at the $n^{-1/4}$ rate, then the FD-DR-Learner $\hat{\tau}_{\mathrm{DR}}$ converges at the quasi-oracle rate (behaving as if the true nuisances were known).

## 4   FD-R-LEARNER

In this section, we introduce the FD-R-Learner, a learner for the conditional FD effect $\tau(C)$ that exhibits a robustness property such that the estimator converges fast even when nuisances converge slowly. To derive the FD-R-Learner, we re-express the data-generating process as the following partial linear model:

---

**Proposition 2** (**Partial Linear Model for FD**). Suppose the FD criterion holds (i.e., Fig. 1a).

$$X = e_X(C) + \epsilon_X, \quad \mathbb{E}[\epsilon_X \mid C] = 0, \tag{29}$$
$$Z = a(C) + Xb(C) + \epsilon_Z, \quad \mathbb{E}[\epsilon_Z \mid XC] = 0, \tag{30}$$
$$Y = f(XC) + Zg(XC) + \epsilon_Y, \quad \mathbb{E}[\epsilon_Y \mid ZXC] = 0, \tag{31}$$

where $e_X(C) \triangleq \mathbb{E}[X \mid C]$ and

$$a(C) \triangleq \mathbb{E}[Z \mid \mathrm{do}(X = 0), C], \quad b(C) \triangleq \mathbb{E}[Z \mid \mathrm{do}(X = 1), C] - a(C),$$
$$f(XC) \triangleq \mathbb{E}[Y \mid \mathrm{do}(Z = 0), XC], \quad g(XC) \triangleq \mathbb{E}[Y \mid \mathrm{do}(Z = 1), XC] - f(XC).$$

---

Since $C$ satisfies the BD criterion relative to $(X, Z)$ (by the definition of the FD-criterion in §2), we can learn $b(C)$ using the BD-R-Learner as follows:

$$\widehat{b} \in \arg\min_{b \in \mathcal{B}} \widehat{L}^{\mathrm{BDR}}_{\lambda_b, \mathcal{D}_2(Z, X, C)}(b, \widehat{\eta}_b \triangleq \{\widehat{e}_X, \widehat{m}_Z\}), \tag{32}$$

where the population and empirical R-risk functions are defined in Eq. (10) and (11), $(\mathcal{D}_1, \mathcal{D}_2)$ is a partition of the sample $\mathcal{D}$, and $\widehat{e}_X, \widehat{m}_Z$ are estimated nuisances for $e_X \triangleq \Pr(X = 1 \mid C)$ and $m_Z \triangleq \mathbb{E}[Z \mid C]$, learned from $\mathcal{D}_1$. error analysis of the BD-R-Learner implies that $\widehat{b}$ converges to $b$ at the quasi-oracle rate even when the estimated nuisances converge as slowly as the $n^{-1/4}$ rate.

Similarly, $g$ can be estimated using the BD-R-Learner, since $(X, C)$ satisfies the BD criterion relative to $(Z, Y)$ (by the definition of the FD-criterion), and $Z$ and $Y$ admit the Robinson (1988)'s partial linear model as in Eq. (7) as follows:

$$Z = e_Z(XC) + \epsilon_Z, \qquad \mathbb{E}[\epsilon_Z \mid XC] = 0, \tag{33}$$
$$Y = f(XC) + Zg(XC) + \epsilon_Y, \quad \mathbb{E}[\epsilon_Y \mid ZXC] = 0, \tag{34}$$

where $e_Z(XC) \triangleq a(C) + Xb(C)$ from Eq. (30), so that $e_Z(XC) = \mathbb{E}[Z \mid XC]$ because $\mathbb{E}[Z \mid \mathrm{do}(X = \bar{x}), C] = \mathbb{E}[Z \mid \bar{x}, C]$. Therefore, $g(XC)$ can be estimated using BD-R-Learner as follows:

$$\widehat{g} \in \arg\min_{g \in \mathcal{Q}} \widehat{L}^{\mathrm{BDR}}_{\lambda_g, \mathcal{D}_2(Y, Z, XC)}(g, \widehat{\eta}_g \triangleq \{\widehat{e}_Z, \widehat{m}_Y\}), \tag{35}$$

where $(\mathcal{D}_{1g}, \mathcal{D}_{2g})$ is a partition of the sample $\mathcal{D}$, and $\widehat{e}_Z, \widehat{m}_Y$ are estimated nuisances for $e_Z \triangleq \mathbb{E}[Z \mid XC]$ and $m_Y \triangleq \mathbb{E}[Y \mid XC]$, learned from $\mathcal{D}_{1g}$. The error analysis of the BD-R-Learner implies that $\widehat{g}$ converges to $g$ at the quasi-oracle rate even when estimated nuisances converge slowly.

Such debiasedness properties of $b(C)$ and $g(XC)$ is crucial for the estimation of heterogeneous FD causal effect $\tau(C)$, since it can be expressed as a functional of $b$ and $g$:

---

**Theorem 2** (**Heterogeneous Treatment Effect via Partial Linear Equation**). Let $b$ and $g$ denote the functionals in Prop. 2. Define $\gamma_g(C) \triangleq \mathbb{E}[g(XC) \mid C]$. Then,

$$\tau(C) = b(C)\gamma_g(C). \tag{36}$$

---

A remaining challenge is to estimate $\gamma_g(C) = e_X(C)g(1C) + (1 - e_X(C))g(0C)$ in a sample-efficient manner. One straightforward approach is to construct a plug-in-estimator by substituting the estimated nuisances:

$$\widehat{\gamma}^{\mathrm{plug}}(C) \triangleq \widehat{e}_X(C)\widehat{g}(1C) + \{1 - \widehat{e}_X(C)\}\widehat{g}(0C). \tag{37}$$

However, even if $\hat{b}$ and $\hat{g}$ converge quickly (Prop. 1), $\widehat{g}^{\mathrm{plug}}$ may still converge slowly when nuisance estimates converge poorly, because its error depends directly on the accuracy of $\widehat{e}_X(C)$. Specifically,

$$\widehat{\gamma}^{\mathrm{plug}}(C) - \gamma_g(C) = \{g(1C) - g(0C)\}(\hat{e}_X(C) - e_X(C)) + \hat{e}_X(C)\{\hat{g}(1C) - g(1C)\} \quad (38)$$
$$+ (1 - \hat{e}_X(C))\{\hat{g}(0C) - g(0C)\}. \quad (39)$$

Thus, the convergence of $\widehat{\gamma}^{\mathrm{plug}}(C)$ is bottlenecked by the accuracy of $\widehat{e}_X$, and can be slow when nuisance estimates converge poorly.

To address this challenge, we define the *pseudo-g function*, which serves as a pseudo-outcome for $g(XC)$, as follows:

---

**Definition 4** (**Pseudo-$g$**). The *pseudo-$g$* $\zeta_{\widetilde{\eta}_z}(XC)$ with $\widetilde{\eta}_z \triangleq \{\tilde{e}_X(C), \tilde{g}(XC)\}$ is:

$$\zeta_{\widetilde{\eta}_z}(XC) \triangleq \{1 - \widetilde{e}_X(C)\}\widetilde{g}(0C) + \widetilde{e}_X(C)\widetilde{g}(1C) + \{X - \widetilde{e}_X(C)\}\{\widetilde{g}(1C) - \widetilde{g}(0C)\}. \quad (40)$$

---

The pseudo-$g$ function enjoys the following consistency and robustness properties:

---

**Lemma 3** (**Property of pseudo-$g$**). Let $\zeta_{\widetilde{\eta}_z}(XC)$ denote the pseudo-$g$ function.

1. **Consistency:** $\mathbb{E}[\zeta_{\eta_z}(XC) \mid C] = \gamma_g(C)$ with $\eta_z \triangleq \{e_X(C), g(XC)\}$.
2. **Error Correction:** For any estimated $\widehat{e}_X$ and $\widehat{g}$,

$$\mathbb{E}[\zeta_{\widehat{\eta}_z}(XC) \mid C] - \gamma_g(C) = e_X(C)\{\widehat{g}(1C) - g(1C)\} + \{1 - e_X(C)\}\{\widehat{g}(0C) - g(0C)\}. \quad (41)$$

---

The error analysis in Lemma 3 implies that the pseudo-$g$'s error no longer depends on the error of $\widehat{e}_X$; it depends only on the error of $\widehat{g}$, which can be learned sample-efficiently via the BD-R-Learner.

We now formally define the FD-R-Learner as follows:

---

**Definition 5** (**FD-R-Learner**). Let $(\mathcal{D}_1, \mathcal{D}_2, \mathcal{D}_3)$ be disjoint (independent) splits of size $n$ of $V_i = (C_i, X_i, Z_i, Y_i)$. Let $\mathcal{B}, \mathcal{Q}, \Gamma$ denote the function classes for $b, g, \gamma$, respectively. Let $\lambda_{b,n}, \lambda_{g,n}, \lambda_{\gamma,n}$ be regularization levels.

1. (**Nuisance Fit**) Fit $\widehat{\eta}_b \triangleq \{\widehat{e}_X, \widehat{m}_Z\}$ and $\widehat{\eta}_g \triangleq \{\widehat{e}_Z, \widehat{m}_Y\}$ using $\mathcal{D}_1$.
2. (**BD-R-Learner for $b$**) Learn $\widehat{b}$ using $\mathcal{D}_2$ with $\widehat{\eta}_b$ from Eq. (32).
3. (**BD-R-Learner for $g$**) Learn $\widehat{g}$ using $\mathcal{D}_2$ with $\widehat{\eta}_g$ from Eq. (35).
4. (**Pseudo-$g$**) Evaluate the pseudo-$g$ function $\zeta_{\widehat{\eta}_z}(XC)$ using $\mathcal{D}_3$.
5. (**Learn $\gamma$**) Find

$$\widehat{\gamma} \in \arg\min_{\gamma \in \Gamma} \frac{1}{|\mathcal{D}_3|} \sum_{i: V_i \in \mathcal{D}_3} \{\zeta_{\widehat{\eta}_z}(X_i C_i) - \gamma(C_i)\}^2 + \lambda_{\zeta,n}\mathcal{J}(\gamma). \quad (42)$$

6. Return $\widehat{\tau}_{\mathrm{R}}(C) \triangleq \widehat{b}(C)\widehat{\gamma}(C)$.

---

One may repeat steps (1-5) with the partitions swapped and average the estimates. Finally, the following result formalizes the robustness and sample-efficiency of the learner:

---

**Theorem 3** (**Error-Analysis of FD-R-Learner**). Let $\mathcal{L}_{\lambda}^{\gamma}(\widetilde{\gamma}, \widetilde{\eta}_z) \triangleq \mathbb{E}[\{\zeta_{\widetilde{\eta}_z}(XC) - \widetilde{\gamma}(C)\}^2] + \lambda\mathcal{J}(\widetilde{\gamma})$ be a population risk for $\gamma$. Suppose there exists quasi-oracle rate

---

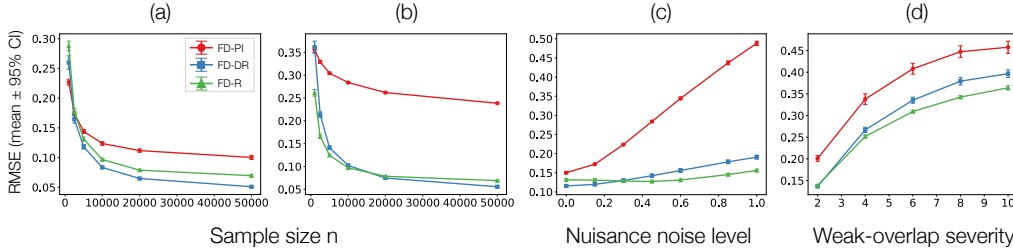

Figure 2: Synthetic study: RMSE (mean $\pm$ 95% CI) of FD-PI (plug-in), FD-DR, and FD-R estimators: **(a–b)** varying sample size $n$, where (a) no structural nuisance noises is added, and (b) nuisances converge at the $n^{-1/4}$ rate; **(c)** varying the level $\rho$ of nuisance noises $\rho\epsilon, \epsilon \sim \mathcal{N}(n^{-1/4}, n^{-1/4})$; and **(d)** varying weak-overlap severity.

---

functions $\mathcal{R}_{\mathcal{B}}(\epsilon, \widetilde{b}, \widetilde{\eta}_b)$, $\mathcal{R}_{\mathcal{Q}}(\epsilon, \widetilde{g}, \widetilde{\eta}_g)$ and $\mathcal{R}_{\Gamma}(\epsilon, \widetilde{\gamma}, \widetilde{\eta}_z)$ such that, with probability $1 - \epsilon$,

$$|L_\lambda^{\mathrm{BDR}}(\widetilde{b}, \widetilde{\eta}_b) - \min_{b' \in \mathcal{B}} L_\lambda(b', \widetilde{\eta}_b)| \leq \mathcal{R}_{\mathcal{B}}(\epsilon, \widetilde{b}, \widetilde{\eta}_b), \tag{43}$$

$$|L_\lambda^{\mathrm{BDR}}(\widetilde{g}, \widetilde{\eta}_g) - \min_{g' \in \mathcal{Q}} L_\lambda(g', \widetilde{\eta}_g)| \leq \mathcal{R}_{\mathcal{Q}}(\epsilon, \widetilde{g}, \widetilde{\eta}_g), \tag{44}$$

$$|\mathcal{L}_\lambda^\gamma(\widetilde{\gamma}, \widetilde{\eta}_z) - \min_{\gamma' \in \Gamma} \mathcal{L}_\lambda^\gamma(\gamma', \widetilde{\eta}_z)| \leq \mathcal{R}_{\Gamma}(\epsilon, \widetilde{\gamma}, \widetilde{\eta}_z). \tag{45}$$

Let $\widehat{\tau}_{\mathrm{R}}$ denote the FD-R-Learner in Def. 5. With probability $1 - \epsilon$,

$$\|\widehat{\tau}_{\mathrm{R}} - \tau\|_2^2 \lesssim \mathcal{R}_{\mathcal{B}}(\epsilon, \widehat{b}, \widehat{\eta}_b) + \mathcal{R}_{\mathcal{Q}}(\epsilon, \widehat{q}, \widehat{\eta}_g) + \mathcal{R}_{\Gamma}(\epsilon, \widehat{\gamma}, \widehat{\eta}_z) \tag{46}$$

$$+ \|\widehat{e}_X - e_X\|_4^4 + \|\widehat{e}_X - e_X\|_4^2 \|\widehat{m}_Z - m_Z\|_4^2 \tag{47}$$

$$+ \|\widehat{e}_Z - e_Z\|_4^4 + \|\widehat{e}_Z - e_Z\|_4^2 \|\widehat{m}_Y - m_Y\|_4^2. \tag{48}$$

---

In words, the error of $\widehat{\tau}_{\mathrm{R}}$ is controlled by the quasi-oracle rates, together with second-order products and fourth-order terms of nuisance errors, reflecting the method's debiasedness. For example, if all nuisances achieve the $n^{-1/4}$ rate, then $\widehat{\tau}_{\mathrm{R}}$ converges at the quasi-oracle rate.

### 4.1 COMPARISON BETWEEN FD-DR-LEARNER AND FD-R-LEARNER

**FD-DR-Learner.** FD-DR enjoys a *double robustness* property (Thm. 1), where its error depends on the quasi-oracle rate plus nuisance errors represented as second-order *products*. Consequently, quasi-oracle rates are attained if either (i) $\widehat{q}$ is accurate, or (ii) $(\widehat{m}, \widehat{e})$ are accurate, even when the other blocks are *misspecified* or converge slowly. This makes FD-DR a default when $\widehat{q}$ or $(\widehat{m}, \widehat{e})$ can be fit well. A caveat is variance under weak overlap (positivity): its construction uses inverse weights and density ratios (e.g., $\pi_{\bar{x}}(XC)$ and $\xi_{\bar{x}}(ZXC)$), which can inflate finite-sample variance when $e$ or $q$ approach 0 or 1.

**FD-R-Learner.** FD-R avoids the density ratios required by FD-DR, making it more variance-friendly under near-violations of overlap. This makes FD-R a strong choice when overlap is weak. In addition, FD-R provides interpretability by decomposing the estimation into two BD-R-learner subproblems for the pathway components $b(C)$ (effect of $X \to Z$) and $g(XC)$ (effect of $Z \to Y$). These intermediates ($b$ and $g$) are useful for diagnostics. However, FD-R requires more nuisance fits ($e_X, m_Z, e_Z, m_Y$ plus $b, g$) compared to the FD-DR-Learner.

**Practitioner Guidance.** Prefer FD-DR-Learner when $\widehat{q}$ or $(\widehat{m}, \widehat{e})$ can be accurately estimated. Prefer FD-R-Learner when overlap is weak (to avoid density ratio estimation) or when interpretability of the $X \to Z$ and $Z \to Y$ pathways is valuable.

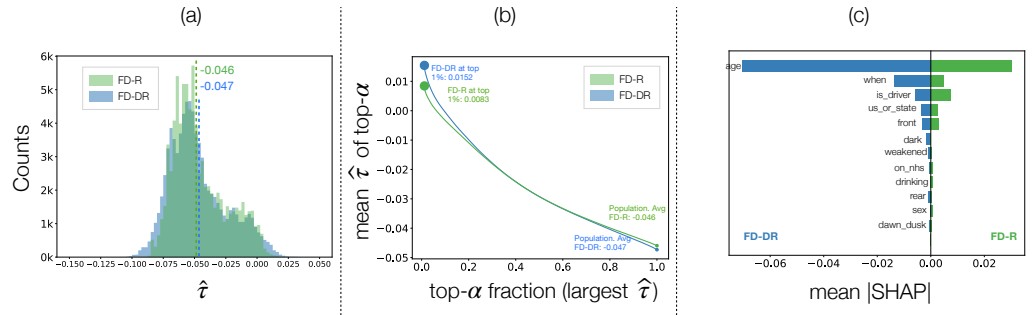

Figure 3: Seat-Belt Laws and Fatalities: **(a)** histograms for $\widehat{\tau}_{DR}(C_i)$ and $\widehat{\tau}_R(C_i)$ **(b)** concentration curve showing the mean values of $\widehat{\tau}_{DR}$ and $\widehat{\tau}_R$ within the top-$\alpha$ fraction (largest $\widehat{\tau}$), where both learners exhibit a downward trend; **(c)** SHAP feature importance highlighting age, time of date, and driver status as the most influential features for both learners.

## 5 EXPERIMENTS

In this section, we demonstrate the debiasedness of the proposed estimators. In all experiments, nuisance functions are learned with XGBoost (Chen & Guestrin, 2016). We compare the proposed FD-DR and FD-R-learners with a plug-in (PI) estimator $\widehat{\tau}_{PI}$ of the target estimand:

$$\widehat{\tau}_{PI}(C) \triangleq \sum_{zx} \{\widehat{q}(z \mid 1C) - \widehat{q}(z \mid 0C)\}\widehat{e}(X = x \mid C)\widehat{m}(zxC). \tag{49}$$

Details of simulations are described in Sec. C. The implementation is available at https://github.com/yonghanjung/FD-CATE.

### 5.1 SYNTHETIC DATA ANALYSIS

We assess the proposed estimators on synthetic scenarios where the true heterogeneous FD effect $\tau(C)$ is known. Fig. 2 reports the root mean squared error (RMSE; mean $\pm$ 95% CI) of the plug-in baseline (FD-PI, $\widehat{\tau}_{PI}$), the FD-DR learner, and the FD-R learner across four regimes.

Panels (a–b) vary the sample size $n$. Both FD-DR and FD-R consistently dominate FD-PI as $n$ grows. When no additional structural noise is introduced (panel (a)), all estimators converge as expected, but the proposed FD-DR and FD-R learners achieve substantially lower error. When the nuisance functions are restricted to converge at the $n^{-1/4}$ rate (panel (b)), FD-DR and FD-R still exhibit reliable convergence, demonstrating robustness to imperfect nuisance estimation, whereas the plug-in estimator converges much more slowly.

To further probe robustness under noisy nuisances, we inject controlled noise of the form $\rho\epsilon$ into the nuisance functionals, where $\epsilon \sim \mathcal{N}(n^{-1/4}, n^{-1/4})$ and $\rho \in \{0.0, 0.2, 0.4, 0.6, 0.8, 1.0\}$. Panel (c) shows that the RMSE of $\widehat{\tau}_{PI}$ deteriorates rapidly as $\rho$ increases, while FD-DR and FD-R maintain stable performance. FD-R achieves lower RMSE across higher noise levels, reflecting its reduced reliance on nuisance accuracy.

Finally, panel (d) examines weak-overlap scenarios by pushing $e(X = 1 \mid C)$ and $q(Z = 1 \mid X, C)$ toward zero. In this regime, the plug-in estimator exhibits severe degradation, and FD-DR suffers noticeable variance inflation due to its use of inverse weights. FD-R, by contrast, remains stable and consistently outperforms both FD-PI and FD-DR. These patterns mirror our analytical comparison in § 4.1, confirming that FD-DR excels when nuisances are accurately estimated and overlap is sufficient, while FD-R is particularly advantageous under weak overlap or noisy nuisance models.

## 5.2 Case Study: State Seat-Belt Laws and Fatalities (FARS)

We use a state–year panel constructed from National Highway Traffic Safety Administration (NHTSA) sources (National Highway Traffic Safety Administration, 2000) using the Fatality Analysis Reporting System (FARS). In our data, the treatment is whether a state–year has a *primary seat-belt law* (enforcing seat-belt use) ($X$), the mediator is the *belt-use* ($Z$) from NHTSA surveys, the outcome is the *occupant fatality $Y$*, and $C$ collects covariates affecting $(X, Z, Y)$. The front-door structure is plausible here because the effect from $X$ to $Y$ operates through increased belt use $Z$; rich set of covariates $C$ helps mitigate confounding bias between $(X, Z)$ and between $(Z, Y)$. Also, positivity holds in this dataset, because belt use is neither zero nor universal across law regimes; i.e., some occupants do not fasten seat belts even under a primary law.

Fig. 3 summarizes the analysis. Panel (a) shows the distributions of $\widehat{\tau}_{\mathrm{DR}}(C_i)$ and $\widehat{\tau}_{\mathrm{R}}(C_i)$, indicating lower fatality rates under primary-law regimes. Panel (b) reports the *concentration curve* (mean $\widehat{\tau}$ within the top-$\alpha$ fraction ranked by $\widehat{\tau}$). Only a small portion of units exhibit increases fatality under the primary law regime ($X{=}1$), whereas over 95% of units show decreases, consistent with a preventive effect of primary-law adoption. Panel (c) presents SHAP feature importance, highlighting age, time of day, and driver status as dominant factors explaining heterogeneity.

Together, our results indicate that primary seat-belt laws reduce occupant fatality rates for the majority of units, illustrating the practical utility of our approach for estimating causal effects from real-world datasets that fit the FD setting.

## 6 Conclusion and Discussion

**Summary**. We developed two heterogeneous FD treatment effects estimators: FD-DR-Learner and FD-R-Learner. Both attain *quasi-oracle* rates under $n^{-1/4}$-rate nuisance convergence (Thm. 1, 3). A comparison of the two estimators for practitioners is given in § 4.1. In synthetic stress tests (varying $n$, slow nuisances, weak overlap) they dominate a plug-in baseline, and in our FARS seat-belt case study they deliver reliable personalized FD estimates.

**Limitations & future work.** (i) *Positivity.* Our guarantees assume overlap for $e(X \mid C)$ and $q(Z \mid X, C)$; near-violations inflate variance (especially for FD-DR). We recommend overlap diagnostics, ratio stabilization, and overlap-aware uncertainty, and plan adaptive routing toward FD-R under weak overlap. (ii) *Binary mediator.* Our theory uses a binary $Z$, whereas many practical settings feature continuous or multidimensional mediators. Extending FD-DR and FD-R learners to handle such settings is a promising direction.

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

# Supplement to "Debiased Front-Door Learners for Heterogeneous Effects"

## CONTENTS

# A    Preliminaries: R-Learner and DR-Learner Analysis using Orthogonal Statistical Learning

We study a population risk $L(\tau, \eta)$, where the *target* $\tau \in \mathcal{T}$ and the *nuisance* $\eta \in \mathcal{H}$ live in normed spaces $(\mathcal{T}, \|\cdot\|_{\mathcal{T}})$ and $(\mathcal{H}, \|\cdot\|_{\mathcal{H}})$, respectively. Throughout, $\eta_0$ denotes the true nuisance. We define the (possibly non-unique) *oracle minimizer*

$$\tau_0 \in \arg\min_{\tau \in \mathcal{T}} L(\tau, \eta_0), \tag{A.1}$$

which we assume is nonempty.

**Directional derivatives.**    For a functional $F$ and direction $h$, the (Gâteaux) derivative with respect to a variable $x$ at $x_0$ is

$$\nabla_x F(x_0)[h] \triangleq \lim_{t \to 0} \frac{F(x_0 + th) - F(x_0)}{t}, \tag{A.2}$$

and second derivatives $\nabla_x^2 F(x_0)[h_1, h_2]$ are defined analogously; mixed derivatives such as $\nabla_\eta \nabla_\tau L$ will be used for orthogonality.

**Sample splitting and plug-in.**    We assume a two-way split into independent folds of approximately equal size: one to learn $\hat{\eta}$ (using data $\mathcal{D}_\eta$), and one to learn $\hat{\tau}$ by minimizing $L(\tau, \hat{\eta})$ over $\tau$, i.e.,

$$\tau_{\hat{\eta}}^* \triangleq \arg\min_\tau L(\tau, \hat{\eta}), \qquad \text{so that} \qquad \tau_0 = \tau_{\eta_0}^*.$$

This separation prevents overfitting-induced bias when we later linearize around $(\tau_0, \eta_0)$.

**Target-class statistical term.**    Let $R_{\mathcal{T}}(\tau; \eta, \epsilon) \geq 0$ be a data-dependent rate function such that, with probability at least $1 - \epsilon$,

$$L(\tau, \eta) - L(\tau_\eta^*, \eta) \leq R_{\mathcal{T}}(\tau; \eta, \epsilon). \tag{A.3}$$

You may instantiate $R_{\mathcal{T}}$ via localized complexity (e.g., critical radius) or algorithm-specific bounds; we keep it abstract to highlight how nuisance error propagates into target error.

**Goal and norms.**    Our goal is to upper bound the *target error* $\|\tau - \tau_0\|_{\mathcal{T}}^2$. When we write $\|\cdot\|_p$ we mean the $L_p(P)$ norm with respect to the underlying distribution.

## A.1    Examples (R- and DR-learners)

We use standard notation: $T \in \{0, 1\}$ (treatment), $X$ (covariates), $Y$ (outcome). The estimand is the CATE

$$\tau_0(X) \triangleq \mathbb{E}[Y(1) - Y(0) \mid X], \tag{A.4}$$

under the usual *positivity* ($c \leq \pi_0(X) \leq 1 - c$ a.s.) and i.i.d. sampling. We assume

$$Y(t) \perp\!\!\!\perp T \mid X \implies \mathbb{E}[Y(t) \mid X] = \mathbb{E}[Y \mid t, X], \ \forall t \in \{0, 1\}. \tag{A.5}$$

### A.1.1    R-Learner

The Robinson decomposition posits

$$Y = f_0(X) + T\,\tau_0(X) + \epsilon_Y, \quad \mathbb{E}[\epsilon_Y \mid T, X] = 0, \tag{A.6}$$

$$T = \pi_0(X) + \epsilon_X, \quad \mathbb{E}[\epsilon_X \mid X] = 0, \tag{A.7}$$

and with $m_0(X) \triangleq \mathbb{E}[Y \mid X]$ we have $m_0(X) = f_0(X) + \pi_0(X)\tau_0(X)$. Hence

$$Y - m_0(X) = \big(T - \pi_0(X)\big)\tau_0(X) + \epsilon_Y. \tag{A.8}$$

Thus, viewing $\tau_0$ as an OLS-type coefficient in a residualized regression, we define

$$L_{\mathrm{R}}(\tau, \eta_0 \triangleq \{m_0, \pi_0\}) \triangleq \mathbb{E}\Big[\big\{Y - m_0(X) - \big(T - \pi_0(X)\big)\tau(X)\big\}^2\Big], \tag{A.9}$$

so that $\tau_0 \in \arg\min_\tau L_{\mathrm{R}}(\tau, \eta_0)$.

### A.1.2 DR-Learner

We define following nuisances:

$$\mu_0(T, X) \triangleq \mathbb{E}[Y \mid T, X], \qquad \omega_0(T, X) \triangleq \frac{2T - 1}{P(T \mid X)}. \tag{A.10}$$

Define the pseudo-outcome

$$\varphi(V; \eta_0 \triangleq \{\mu_0, \pi_0\}) \triangleq \omega_0(T, X)\{Y - \mu_0(T, X)\} + \mu_0(1, X) - \mu_0(0, X), \tag{A.11}$$

and the squared-loss objective

$$L_{\mathrm{DR}}(\tau, \eta) \triangleq \mathbb{E}\Big[\big\{\varphi(V; \eta) - \tau(X)\big\}^2\Big]. \tag{A.12}$$

This loss is centered at the CATE in virtue of $\mathbb{E}[\varphi(V; \eta_0) \mid X] = \tau_0(X)$.

### A.2 Assumptions

We now state structural conditions that yield fast rates. The exposition follows the orthogonal-statistical-learning (OSL) template: first-order optimality at truth, curvature in $\tau$, and orthogonality to damp the impact of nuisance error.

---

**Assumption A.1** (**First-order optimality in** $\tau$). Moving away from $\tau_0$ cannot reduce the population risk at the true nuisance:

$$\nabla_\tau L(\tau_0, \eta_0)[h_\tau] \geq 0 \quad \text{for all feasible directions } h_\tau \text{ from } \tau_0. \tag{A.13}$$

---

**Assumption A.2** (**Strong convexity (quadratic growth) in** $\tau$). There exist constants $\lambda > 0$, $\kappa \geq 0$, and $r \in [0, 1)$ such that, for any $\bar{\tau}$ on the line segment between $\tau$ and $\tau_0$,

$$\nabla_\tau^2 L(\bar{\tau}, \eta)[\tau - \tau_0, \tau - \tau_0] \geq \lambda\|\tau - \tau_0\|_{\mathcal{T}}^2 - \kappa\|\eta - \eta_0\|_{\mathcal{H}}^{\frac{4}{1+r}}. \tag{A.14}$$

*Rationale:* The risk function $L(\tau, \eta)$ is in a bowl-shape over $\tau$. The $\kappa$ term allows mild curvature deterioration when $\eta \neq \eta_0$; the exponent $4/(1+r)$ is chosen to balance mixed terms via Young's inequality later.

---

### A.3 Assumption checks for the examples

We verify that the R- and DR-losses satisfy the above, clarifying how positivity yields curvature and how residualization/DR construction yields orthogonality.

### A.3.1 R-Learner: assumptions hold

**First-order optimality.** With $\tilde{Y} \triangleq Y - m_0(X)$, $\tilde{T} \triangleq T - \pi_0(X)$,

$$\nabla_\tau L_{\mathrm{R}}(\tau_0, \eta_0)[h_\tau] = -2\,\mathbb{E}\big[(\tilde{Y} - \tilde{T}\tau_0)\,\tilde{T}\,h_\tau(X)\big] = -2\,\mathbb{E}\big[\mathbb{E}[\epsilon_Y \mid T, X]\tilde{T}\,h_\tau(X)\big] = 0. \tag{A.15}$$

Hence Assumption A.1 holds.

**Strong convexity.** We have

$$\nabla_\tau^2 L(\bar\tau, \eta)[\tau - \tau_0, \tau - \tau_0] = 2\mathbb{E}\big[\{\tau(X) - \tau_0(X)\}^2 \{T - \pi(X)\}^2\big], \qquad (A.16)$$

where

$$\mathbb{E}\big[(T - \pi(X))^2 \mid X\big] = \underbrace{\mathrm{Var}(T \mid X)}_{=\pi_0(X)(1-\pi_0(X))} + \big(\pi_0(X) - \pi(X)\big)^2 \geq \pi_0(X)(1 - \pi_0(X)). \quad (A.17)$$

Therefore,

$$\nabla_\tau^2 L(\bar\tau, \eta)[\tau - \tau_0, \tau - \tau_0] = 2\mathbb{E}\big[\{\tau(X) - \tau_0(X)\}^2 \{T - \pi(X)\}^2\big] \qquad (A.18)$$

$$\geq 2\mathbb{E}\big[\{\tau(X) - \tau_0(X)\}^2 \mathrm{Var}(T \mid X)\big] \qquad (A.19)$$

$$= 2\mathbb{E}\big[\{\tau(X) - \tau_0(X)\}^2 \pi_0(X)\{1 - \pi_0(X)\}\big] \qquad (A.20)$$

$$\geq 2E\big[\{\tau(X) - \tau_0(X)\}^2 c\{1 - c\}\big] \qquad (A.21)$$

$$= 2c(1 - c)\|\tau - \tau_0\|_2^2. \qquad (A.22)$$

Hence, Assumption A.2 holds with $\lambda = 2c(1 - c)$ and $\kappa = 0$ (taking $\|\cdot\|_\mathcal{T} = \|\cdot\|_2$).

### A.3.2 DR-Learner: assumptions hold

**First-order optimality.**
$$\nabla_\tau L_{\mathrm{DR}}(\tau_0, \eta_0)[h_\tau] = -2\,\mathbb{E}[\{\varphi(V; \eta_0) - \tau_0(X)\}\,h_\tau(X)] = 0, \qquad (A.23)$$

since $\mathbb{E}[\varphi(V; \eta_0) \mid X] = \tau_0(X)$.

**Strong convexity.** We first note that

$$\nabla_\tau L_{\mathrm{DR}}(\tau_0, \eta_0)[\tau - \tau_0] = -2\mathbb{E}[\{\varphi(V; \eta) - \tau(X)\}\{\tau(X) - \tau_0(X)\}]. \qquad (A.24)$$

This gives

$$\nabla_\tau^2 L_{\mathrm{DR}}(\tau, \eta)[\tau - \tau_0, \tau - \tau_0] = 2\|\tau - \tau_0\|_2^2, \qquad (A.25)$$

which shows that $\kappa = 0$ and $\lambda = 2$ (taking $\|\cdot\|_\mathcal{T} = \|\cdot\|_2$).

### A.4 Main Result

---

**Theorem A.1** (**Fast Rate Convergence**). Suppose Assumption A.1 and A.2 hold. Then,

$$\|\tau - \tau_0\|_\mathcal{T}^2 \leq \tfrac{2}{\lambda} R_\mathcal{T}(\hat\tau; \hat\eta, \epsilon) + \tfrac{2}{\lambda}\{\nabla_\tau L(\tau_0, \eta_0)[\hat\tau - \tau_0] - \nabla_\tau L(\tau_0, \hat\eta)[\hat\tau - \tau_0]\} \quad (A.26)$$

$$+ \tfrac{\kappa}{\lambda}\|\eta - \eta_0\|_\mathcal{H}^{\frac{4}{1+r}}. \qquad (A.27)$$

---

**Proof of Thm. A.1.** By applying the Taylor's expansion and rearranging, we have

$$\tfrac{1}{2}\nabla_\tau^2 L(\bar\tau, \hat\eta)[(\hat\tau - \tau_0)^2] = L(\hat\tau, \hat\eta) - L(\tau_0, \hat\eta) - \nabla_\tau L(\tau_0, \hat\tau)[\hat\tau - \tau_0],$$

where $\bar\tau$ is on the line segment between $\hat\tau$ and $\tau_0$.
Using Assumption A.2, we have

$$\frac{\lambda}{2}\|\tau - \tau_0\|_\mathcal{T}^2 \leq \underbrace{L(\hat\tau, \hat\eta) - L(\tau_0, \hat\eta)}_{R_\mathcal{T}(\hat\tau;\hat\eta,\epsilon)} - \nabla_\tau L(\tau_0, \hat\tau)[\hat\tau - \tau_0] + \frac{\kappa}{2}\|\eta - \eta_0\|_\mathcal{H}^{\frac{4}{1+r}}.$$

---

Since $\nabla_\tau L(\tau_0, \eta_0)[\hat{\tau} - \tau_0] \geq 0$ by Assumption A.1, we have

$$\frac{\lambda}{2}\|\tau - \tau_0\|_{\mathcal{T}}^2 \leq R_{\mathcal{T}}(\hat{\tau}; \hat{\eta}, \epsilon) + \{\nabla_\tau L(\tau_0, \eta_0)[\hat{\tau} - \tau_0] - \nabla_\tau L(\tau_0, \hat{\tau})[\hat{\tau} - \tau_0]\} \quad \text{(A.28)}$$

$$+ \frac{\kappa}{2}\|\eta - \eta_0\|_{\mathcal{H}}^{\frac{4}{1+r}}. \quad \text{(A.29)}$$

$\square$

The middle difference

$$\{\nabla_\tau L(\tau_0, \eta_0) - \nabla_\tau L(\tau_0, \hat{\eta})\}[\hat{\tau} - \tau_0] \quad \text{(A.30)}$$

is the *nuisance leakage of the first-order optimality condition.* It is the main channel through which nuisance error affects the target. Under Neyman orthogonality, the leakage is *higher than first order* in $\|\hat{\eta} - \eta_0\|$ (typically quadratic or a product of nuisance errors), so $\hat{\tau}$ inherits only a higher-order remainder rather than linear bias. In particular, for the DR-learner it factors into a product of nuisance errors (yielding *double robustness*), whereas for the R-learner it enables *fast rates* once the nuisances are sufficiently accurate. We quantify these forms below for each loss.

### A.4.1 NUISANCE LEAKAGE: R-LEARNER

**Theorem A.2** (**Error Analysis: R-learner**). Suppose Assumption A.1 and A.2 hold with $\|\cdot\|_{\mathcal{T}} = \|\cdot\|_2$. Let $a \triangleq \|\tau_0\|_\infty^2$ and $\lambda \triangleq 2c(1-c)$, where $c$ is a constant satisfying $c \leq \pi_0(X) \leq 1 - c$. Then, with probability $1 - \epsilon$,

$$\|\hat{\tau} - \tau_0\|_2^2 \leq \frac{4}{\lambda}\mathcal{R}_{\mathcal{T}}(\hat{\tau}; \hat{\eta}, \epsilon) + \frac{32a}{\lambda^2}\|\hat{\pi} - \pi_0\|_4^4 + \frac{32}{\lambda^2}\|\hat{m} - m_0\|_4^2\|\hat{\pi} - \pi_0\|_4^2. \quad \text{(A.31)}$$

***Proof of Thm. A.2.*** Let $h_\tau(X) \triangleq \hat{\tau}(X) - \tau_0(X)$. Let $\delta_m(X) \triangleq (m(X) - m_0(X))$ and $\delta_\pi(X) \triangleq (\pi(X) - \pi_0(X))$.
Note, the first-order risk function is the following:

$$\nabla_\tau L_{\mathrm{R}}[\tau, \eta](h_\tau) = -2\mathbb{E}[\{Y - m(X) - \tau(X)(T - \pi(X))\} \cdot \{T - \pi(X)\} \cdot h_\tau(X)], \quad \text{(A.32)}$$

We note that $\nabla_\tau L_{\mathrm{R}}[\tau_0, \eta_0](h_\tau) = 0$, as shown in the first-order optimality condition analysis. To analyze the leakage, we rewrite a few terms here:

$$Y - m - \tau_0(T - \pi) = \underbrace{Y - m_0 - \tau_0(T - \pi_0)}_{\epsilon_Y} - \delta_m + \tau_0\delta_\pi \quad \text{(A.33)}$$

$$T - \pi = T - \pi_0 - \delta_\pi. \quad \text{(A.34)}$$

Then, we can rewrite the first-order risk as follows:

$$\nabla_\tau L_{\mathrm{R}}[\tau_0, \eta](h_\tau) = -2\mathbb{E}[\{\epsilon_Y - \delta_m + \tau_0\delta_\pi\} \cdot (T - \pi_0 - \delta_\pi) \cdot h_\tau] \quad \text{(A.35)}$$

$$= 2\mathbb{E}[\{\tau_0\delta_\pi^2 - \delta_m\delta_\pi\}h_\tau] \quad \text{(A.36)}$$

$$\leq 2|\mathbb{E}[\tau_0\delta_\pi^2 h_\tau]| + 2|\mathbb{E}[\delta_m\delta_\pi h_\tau]| \quad \text{(A.37)}$$

$$\leq 2\|\delta_\pi\|_4^2 \cdot \|\tau_0\|_\infty \cdot \|h_\tau\|_2 + 2\|\delta_m\|_4 \cdot \|\delta_\pi\|_4 \cdot \|h_\tau\|_2 \quad \text{(A.38)}$$

$$= 2\|h_\tau\|_2 \cdot (\|\tau_0\|_\infty\|\delta_\pi\|_4^2 + \|\delta_m\|_4 \cdot \|\delta_\pi\|_4). \quad \text{(A.39)}$$

Then, for any $\alpha > 0$, Young's inequality (with $p = q = 2$) gives

$$\nabla_\tau L(\tau_0, \eta_0)[\hat{\tau} - \tau_0] - \nabla_\tau L(\tau_0, \hat{\eta})[\hat{\tau} - \tau_0] \tag{A.40}$$

$$\leq 2\|h_\tau\|_2 \cdot \left(\|\tau_0\|_\infty \|\delta_\pi\|_4^2 + \delta_m\|_4 \cdot \|\delta_\pi\|_4\right) \tag{A.41}$$

$$\leq \alpha\|h_\tau\|_2^2 + \frac{1}{\alpha} \left(\|\tau_0\|_\infty \|\delta_\pi\|_4^2 + \|\delta_m\|_4 \cdot \|\delta_\pi\|_4\right)^2 \tag{A.42}$$

$$= \alpha\|h_\tau\|_2^2 + \frac{2}{\alpha}\|\tau_0\|_\infty^2 \|\delta_\pi\|_4^4 + \frac{2}{\alpha}\|\delta_m\|_4^2\|\delta_\pi\|_4^2. \tag{A.43}$$

Choose $\alpha = \lambda/4$. Let $\mathcal{R}_\mathcal{T} \triangleq \mathcal{R}_\mathcal{T}(\hat{\tau}; \hat{\eta}, \epsilon)$. Then, by Thm. A.1, we have

$$\|h_\tau\|_2^2 \leq \frac{2}{\lambda}\mathcal{R}_\mathcal{T} + \frac{2}{\lambda}\frac{\lambda}{4}\|h_\tau\|_2^2 + \frac{16a}{\lambda^2}\|\delta_\pi\|_2^2 + \frac{16}{\lambda^2}\|\delta_m\|_4^2\|\delta_\pi\|_4^2, \tag{A.44}$$

$$\implies \frac{1}{2}\|h_\tau\|_2^2 \leq \frac{2}{\lambda}\mathcal{R}_\mathcal{T} + \frac{16a}{\lambda^2}\|\delta_\pi\|_2^2 + \frac{16}{\lambda^2}\|\delta_m\|_4^2\|\delta_\pi\|_4^2, \tag{A.45}$$

which completes the proof.

$\square$

### A.4.2 NUISANCE LEAKAGE: DR-LEARNER

**Theorem A.3** (**Error Analysis: DR-learner**). Suppose Assumption A.1 and A.2 hold with $\|\cdot\|_\mathcal{T} = \|\cdot\|_2$. Then, with probability $1 - \epsilon$,

$$\|\hat{\tau} - \tau_0\|_2^2 \leq 2\mathcal{R}_\mathcal{T}(\hat{\tau}; \hat{\eta}, \epsilon) + 8\|\omega - \omega_0\|_4^2\|\mu_0 - \mu\|_4^2. \tag{A.46}$$

**Proof of Thm. A.3.** Let $h_\tau(X) \triangleq \hat{\tau}(X) - \tau_0(X)$. Let $\delta_\mu(XZ) \triangleq (\mu(XZ) - \mu_0(XZ))$ and $\delta_\omega(X) \triangleq (\omega(X) - \omega_0(X))$.
Note

$$\nabla_\tau L_{\mathrm{DR}}(\tau, \eta)[h_\tau] = -2\mathbb{E}[\{\varphi(V; \eta) - \tau\}h_\tau] \leq 2|\mathbb{E}[\{\varphi(V; \eta) - \tau\}h_\tau]|. \tag{A.47}$$

We note $\nabla_\tau L_{\mathrm{DR}}(\tau_0, \eta_0)[h_\tau] = 0$, as shown in the first-order optimality condition analysis. Also,

$$|\mathbb{E}[\{\varphi(V; \eta) - \tau_0\}h_\tau]|| \tag{A.48}$$

$$= |\mathbb{E}[\{\omega(TX)\{Y - \mu(TX)\} + \omega_0(TX)\mu(TX) - \omega_0(TX)\mu_0(TX)\}h_\tau(X)]| \tag{A.49}$$

$$= |\mathbb{E}[\{\omega(TX)\{\mu_0(TX) - \mu(TX)\} + \omega_0(TX)\{\mu(TX) - \mu_0(TX)\}\}h_\tau(X)]| \tag{A.50}$$

$$= |\mathbb{E}[\{\omega(TX) - \omega_0(TX)\}\{\mu_0(TX) - \mu(TX)\}h_\tau(X)]| \tag{A.51}$$

$$\leq \|h_\tau\|_2\|(\omega - \omega_0)(\mu_0 - \mu)\|_2 \tag{A.52}$$

$$\leq \|h_\tau\|_2\|\omega - \omega_0\|_4\|\mu_0 - \mu\|_4. \tag{A.53}$$

Then, for any $\alpha > 0$, Young's inequality (with $p = q = 2$) gives

$$2|\mathbb{E}[\{\varphi(V; \eta) - \tau_0\}h_\tau]|| \tag{A.54}$$

$$\leq \alpha\|h_\tau\|_2^2 + \frac{1}{\alpha}\|\omega - \omega_0\|_4^2\|\mu_0 - \mu\|_4^2. \tag{A.55}$$

Choose $\alpha = \lambda/4$. Let $\mathcal{R}_\mathcal{T} \triangleq \mathcal{R}_\mathcal{T}(\hat{\tau}; \hat{\eta}, \epsilon)$. Then, by Thm. A.1, we have

$$\|h_\tau\|_2^2 \leq \frac{2}{\lambda}\mathcal{R}_\mathcal{T} + \frac{2}{\lambda}\frac{\lambda}{4}\|h_\tau\|_2^2 + \frac{16}{\lambda^2}\|\omega - \omega_0\|_4^2\|\mu_0 - \mu\|_4^2 \tag{A.56}$$

$$\implies \frac{1}{2}\|h_\tau\|_2^2 \leq \frac{2}{\lambda}\mathcal{R}_\mathcal{T} + \frac{16}{\lambda^2}\|\omega - \omega_0\|_4^2\|\mu_0 - \mu\|_4^2, \tag{A.57}$$

which completes the proof. □

# B  Proofs

## B.1  Proof of Lemma 1

First,

$$\mathbb{E}[\tau_{\bar{x}}(C)] = \sum_{zxc} m(zxc)q(z \mid \bar{x}c)e(x \mid c)p(c). \tag{B.1}$$

Then,

$$\mathbb{E}[\xi_{\bar{x}}Y] = \sum_{zxc} \mathbb{E}[Y \mid zxc]P(zxc)\frac{q(z \mid \bar{x}c)}{q(z \mid xc)} = \sum_{zxc} m(zxc)q(z \mid \bar{x}c)e(x \mid c)p(c). \tag{B.2}$$

Also,

$$\mathbb{E}[\pi_{\bar{x}}r_{me}] = \sum_{zx'c} \frac{\mathbb{I}_{\bar{x}}(x')}{e(x' \mid c)} \sum_x m(zxc)e(x \mid c)q(z \mid x'c)e(x' \mid c)p(c) \tag{B.3}$$

$$= \sum_{zxc} m(zxc)e(x \mid c)q(z \mid \bar{x}c)p(c). \tag{B.4}$$

Finally,

$$\mathbb{E}[S_{mq_{\bar{x}}}(XC)] = \sum_{xc} S_{mq_{\bar{x}}}(xc)e(x \mid c)p(c) \tag{B.5}$$

$$= \sum_{zxc} m(zxc)q(z \mid \bar{x}c)e(x \mid c)p(c). \tag{B.6}$$

## B.2  Proof of Lemma 2

First,

$$\mathbb{E}[\xi_{\bar{x}}(ZXC)\{Y - m(ZXC)\}] = \mathbb{E}[\xi_{\bar{x}}(ZXC)\{m(ZXC) - m(ZXC)\}] = 0. \tag{B.7}$$

Also,

$$\mathbb{E}[\pi_{\bar{x}}(XC)\{r_{me}(ZC) - \nu_{meq}(XC)\}] = \mathbb{E}[\pi_{\bar{x}}(XC)\{\mathbb{E}[r_{me}(ZC) \mid XC] - \nu_{meq}(XC)\}] = 0, \tag{B.8}$$

since

$$\mathbb{E}[r_{me}(ZC) \mid XC] = \sum_z \underbrace{\sum_x m(zxC)e(x \mid C)}_{=r_{me}(zC)} q(z \mid XC) = \nu_{meq}(XC). \tag{B.9}$$

Finally,

$$\mathbb{E}[s_{mq_{\bar{x}}}(XC) \mid C] = \sum_{zx} m(zxC)q(z \mid \bar{x}C)e(x \mid C) = \tau_{\bar{x}}(C). \tag{B.10}$$

Therefore, $\mathbb{E}[\varphi_{\bar{x}}(V; \eta) \mid C] = \mathbb{E}[s_{mq_{\bar{x}}}(XC) \mid C] = \tau_{\bar{x}}(C)$ .

Now, we prove the doubly robustness. Let

$$T_1(V; \hat{\eta}) \triangleq \widehat{\xi}_{\bar{x}}(ZXC)\{Y - \widehat{m}(ZXC)\} + \widehat{\pi}_{\bar{x}}(XC)\{r_{\hat{m}\hat{e}}(ZC) - \nu_{\hat{m}\hat{q}\hat{e}}(XC)\} + \nu_{\hat{m}\hat{q}\hat{e}}(\bar{x}C)$$

$$T_2(V; \hat{\eta}) \triangleq \sum_{x'} \{\mathbb{I}_{x'}(X) - \hat{e}(x' \mid C)\}s_{\hat{m}\hat{q}}(x'C).$$

We note that

$$T_2(V; \hat\eta) = s_{\hat m\hat q}(XC) - \underbrace{\sum_{x'} \hat e(x' \mid C)\}s_{\hat m\hat q}(x'C)}_{=\nu_{\hat m\hat q\hat e}(\bar x C)}. \tag{B.11}$$

Therefore,

$$T_1(V; \hat\eta) + T_2(V; \hat\eta) \tag{B.12}$$

$$= \widehat{\xi}_{\bar x}(ZXC)\{Y - \widehat m(ZXC)\} + \widehat\pi_{\bar x}(XC)\{r_{\hat m\hat e}(ZC) - \nu_{\hat m\hat q\hat e}(XC)\} \tag{B.13}$$

$$+ \nu_{\hat m\hat q\hat e}(\bar x C) + s_{\hat m\hat q}(XC) - \nu_{\hat m\hat q\hat e}(\bar x C) \tag{B.14}$$

$$= \varphi_{\bar x}(V; \hat\eta). \tag{B.15}$$

Now, from $\mathbb{E}[\tau_{\bar x}(C)] = \mathbb{E}[\xi_{\bar x}(ZXC)m(ZXC)]$,

$$\mathbb{E}[\varphi_{\bar x}(V; \hat\eta) - \tau_{\bar x}(C)]$$

$$= \mathbb{E}[T_1(V; \hat\eta) + T_2(V; \hat\eta) - \xi_{\bar x}(ZXC)m(ZXC)]$$

$$= \mathbb{E}[\widehat{\xi}_{\bar x}(ZXC)\{Y - \widehat m(ZXC)\}] + \mathbb{E}[\xi_{\bar x}(ZXC)\widehat m(ZXC)] - \underbrace{\mathbb{E}[\xi_{\bar x}(ZXC)m(ZXC)]}_{=\mathbb{E}[\tau_{\bar x}(C)]}$$

$$+ \mathbb{E}[\widehat\pi_{\bar x}(XC)\{r_{\hat m\hat e}(ZC) - \nu_{\hat m\hat q\hat e}(XC)\}] + \underbrace{\mathbb{E}[\pi_{\bar x}(XC)\nu_{\hat m\hat q\hat e}(XC)]}_{=\mathbb{E}[\nu_{\hat m\hat q\hat e}(\bar x C)\ ]} - \mathbb{E}[\pi_{\bar x}(XC)\nu_{\hat m q\hat e}(XC)]$$

$$+ \mathbb{E}[\nu_{\hat m q\hat e}(\bar x C) - \xi_{\bar x}(ZXC)\widehat m(ZXC) + s_{\hat m\hat q}(XC) - \nu_{\hat m\hat q\hat e}(\bar x C)].$$

The first line can be handled as follow:

$$\mathbb{E}[\widehat{\xi}_{\bar x}\{Y - \widehat m\} + \xi_{\bar x}\widehat m - \xi_{\bar x}m] = \mathbb{E}[\{\widehat{\xi}_{\bar x} - \xi_{\bar x}\}\{m - \widehat m\}]. \tag{B.16}$$

The second line can be handled as follow:

$$\mathbb{E}[\widehat\pi_{\bar x}\{r_{\hat m\hat e} - \nu_{\hat m\hat q\hat e}\} + \pi_{\bar x}\nu_{\hat m\hat q\hat e} - \pi_{\bar x}\nu_{\hat m q\hat e}] = \mathbb{E}[\{\widehat\pi_{\bar x} - \pi_{\bar x}\}\{\nu_{\hat m q\hat e} - \nu_{\hat m\hat q\hat e}\}]. \tag{B.17}$$

The third line is handled as follow:

$$\mathbb{E}[\nu_{\hat m q\hat e}(\bar x C) - \xi_{\bar x}(ZXC)\widehat m(ZXC) + s_{\hat m\hat q}(XC) - \nu_{\hat m\hat q\hat e}(\bar x C)] \tag{B.18}$$

$$= \sum_{zx} \mathbb{E}[\hat m(zxC)q(z \mid \bar x C)\{\hat e(x \mid C) - e(x \mid C)\}] \tag{B.19}$$

$$+ \sum_{zx} \mathbb{E}[\hat m(zxC)\hat q(z \mid \bar x C)\{e(x \mid C) - \hat e(x \mid C)\}] \tag{B.20}$$

$$= \sum_{zx} \mathbb{E}[\hat m(zxC)\{q(z \mid \bar x C) - \hat q(z \mid \bar x C)\}\{\hat e(x \mid C) - e(x \mid C)\}]. \tag{B.21}$$

This decomposition completes the proof.

## B.3    Proof of Theorem 1

By Thm. A.1, we have

$$\|\widehat\tau^{\mathrm{DR}} - \tau\|_{\mathcal{T}}^2 \le \tfrac{2}{\lambda}R_{\mathcal{T}}(\hat\tau; \hat\eta, \epsilon) + \{\nabla_\tau L_\lambda^{\mathrm{DR}}(\tau, \eta)[\hat\tau - \tau] - \nabla_\tau L_\lambda^{\mathrm{DR}}(\tau, \hat\eta)[\hat\tau - \tau]\}, \tag{B.22}$$

since $\nabla_\tau^2 L_\lambda^{\mathrm{DR}}(\tau, \eta)[\hat\tau - \tau, \hat\tau - \tau] = 2\|\hat\tau - \tau\|_2^2 + \nabla_\tau^2 \mathcal{J}(\tau)$ (i.e., the strong convexity holds).

Define $h_\tau(C) \triangleq \widehat{\tau}(C) - \tau(C)$. First,

$$\begin{aligned}
\nabla_\tau L_\lambda^{\text{DR}}(\tau, \eta)[h_\tau] &= -2\mathbb{E}[\{\varphi_1(V; \eta) - \varphi_0(V; \eta) - \tau_1 + \tau_0\}h_\tau] \\
&= -2\mathbb{E}[\{\varphi_1(V; \eta) - \tau_1(C)\}h_\tau] + 2\mathbb{E}[\{\varphi_0(V; \eta) - \tau_0(C)\}h_\tau] \\
&\lesssim \sum_{\bar{x} \in \{0,1\}} \mathbb{E}[\{\varphi_{\bar{x}}(V; \eta) - \tau_{\bar{x}}(C)\}h_\tau(C)] \\
&= \sum_{\bar{x} \in \{0,1\}} \mathbb{E}_C[\mathbb{E}[\{\varphi_{\bar{x}}(V; \eta) \mid C] - \tau_{\bar{x}}(C)\}h_\tau(C)] \\
&= 0.
\end{aligned}$$

Next,

$$\begin{aligned}
&\nabla_\tau L_\lambda^{\text{DR}}(\tau, \widehat{\eta})[h_\tau] \\
&\lesssim \sum_{\bar{x} \in \{0,1\}} \mathbb{E}[\{\varphi_{\bar{x}}(V; \widehat{\eta}) - \tau_{\bar{x}}(C)\}h_\tau(C)] \\
&= \sum_{\bar{x} \in \{0,1\}} \mathbb{E}_C[\mathbb{E}[\{\varphi_{\bar{x}}(V; \widehat{\eta}) \mid C] - \tau_{\bar{x}}(C)\}h_\tau(C)] \\
&= \sum_{\bar{x} \in \{0,1\}} \mathbb{E}\left[\{\widehat{\xi}_{\bar{x}}(ZXC) - \xi_{\bar{x}}(ZXC)\}\{m(ZXC) - \widehat{m}(ZXC)\}h_\tau(C)\right] \\
&+ \sum_{\bar{x} \in \{0,1\}} \mathbb{E}[\{\widehat{\pi}_{\bar{x}}(XC) - \pi_{\bar{x}}(XC)\}\{\nu_{\hat{m}\hat{q}\hat{e}}(ZXC) - \nu_{\hat{m}\hat{q}\hat{e}}(ZXC)\}h_\tau(C)] \\
&+ \sum_{\bar{x} \in \{0,1\}} \sum_{zx} \mathbb{E}_C\left[\widehat{m}(zxC)\{q(z \mid \bar{x}C) - \widehat{q}(z \mid \bar{x}C)\{\widehat{e}(x \mid C) - e(x \mid C)\}h_\tau(C)\right] \\
&\leq \|h_\tau\|_2 \underbrace{\sum_{\bar{x} \in \{0,1\}} \|\widehat{m} - m\|_4 \|\widehat{\xi}_{\bar{x}} - \xi_{\bar{x}}\|_4}_{A} + \|h_\tau\|_2 \underbrace{\sum_{\bar{x} \in \{0,1\}} \|\nu_{\hat{m}\hat{q}\hat{e}} - \nu_{\hat{m}q\hat{e}}\|_4 \|\widehat{\pi}_{\bar{x}} - \pi_{\bar{x}}\|_4}_{B} \\
&+ \|h_\tau\|_2^2 \underbrace{\sum_{\bar{x} \in \{0,1\}} \sum_{zx} \|\widehat{m}(zxC)\|_\infty \|\widehat{q}(z \mid \bar{x}C) - q(z \mid \bar{x}C)\|_4 \|\widehat{e}(x \mid C) - e(x \mid C)\|_4}_{C} \\
&\leq \alpha \|h_\tau\|_2^2 + \frac{1}{\alpha}(A + B + C), \quad \forall \alpha > 0.
\end{aligned}$$

Choosing $\alpha = \lambda/4$ completes the proof.

## B.4 PROOF OF PROPOSITION 2

Define

$$\epsilon_X \triangleq X - e(C) \tag{B.23}$$

$$\epsilon_Z \triangleq Z - a(C) - Xb(C) \tag{B.24}$$

$$\epsilon_Y \triangleq Y - f(XC) - Zg(XC). \tag{B.25}$$

Then, $\mathbb{E}[\epsilon_X \mid C] = 0$. Next, consider $\epsilon_Z$. For $\bar{x} \in \{0, 1\}$,

$$\mathbb{E}[\epsilon_Z \mid \bar{x}C] = \mathbb{E}[Z \mid \bar{x}C] - a(C) - \bar{x}b(C). \tag{B.26}$$

We note that $\mathbb{E}[\epsilon_Z \mid 1C] = 0$ since $\mathbb{E}[Z \mid X = 1, C] = a(C) + b(C)$. Also, $\mathbb{E}[\epsilon_Z \mid 0C] = 0$ since $\mathbb{E}[Z \mid X = 0, C] = a(C)$ Therefore, $\mathbb{E}[\epsilon_Z \mid XC] = 0$.

Next, consider $\epsilon_Y$.

$$\mathbb{E}[\epsilon_Y \mid Z = 0, XC] = \mathbb{E}[Y \mid Z = 0, XC] - f(XC) \tag{B.27}$$
$$= m(Z = 0, XC) - m(Z = 0, XC) \tag{B.28}$$
$$= 0, \tag{B.29}$$

and

$$\mathbb{E}[\epsilon_Y \mid Z = 1, XC] = \mathbb{E}[Y \mid Z = 1, XC] - f(XC) - g(XC) \tag{B.30}$$
$$= m(Z = 1, XC) - m(Z = 0, XC) - m(Z = 1, XC) + m(Z = 0, XC) \tag{B.31}$$
$$= 0. \tag{B.32}$$

## B.5 Proof of Theorem 2

We first note that $m(ZXC) = f(XC) + Zg(XC)$. Define $\bar{f}(C) \triangleq \sum_x e(x \mid C)f(xC)$ and $\bar{g}(C) \triangleq \sum_x e(x \mid C)g(xC)$ Then,

$$\tau_{\bar{x}}(C) = \mathbb{E}\left[\sum_x e(x \mid C)m(ZxC) \mid \bar{x}C\right] \tag{B.33}$$

$$= \mathbb{E}\left[\sum_x e(x \mid C)\{f(xC) + Zg(xC)\} \mid \bar{x}C\right] \tag{B.34}$$

$$= \mathbb{E}[\bar{f}(C) + Z\bar{g}(C) \mid \bar{x}C]. \tag{B.35}$$

From $\mathbb{E}[Z \mid \bar{x}C] = a(C) + \bar{x}b(C)$,

$$\tau_{\bar{x}}(C) = \mathbb{E}[\bar{f}(C) + Z\bar{g}(C) \mid \bar{x}C] \tag{B.36}$$
$$= \bar{f}(C) + \bar{g}(C)\mathbb{E}[Z \mid \bar{x}C] \tag{B.37}$$
$$= \bar{f}(C) + \bar{g}(C)\{a(C) + \bar{x}b(C)\}. \tag{B.38}$$

This implies that

$$\tau_1(C) - \tau_0(C) = b(C)\bar{g}(C) = b(C)\mathbb{E}[g(XC) \mid C]. \tag{B.39}$$

## B.6 Proof of Lemma 3

We first prove the consistency result:

$$\mathbb{E}[\zeta_{e_X,g}(XC) \mid C] = \{1 - e_X(C)\}g(0C) + e_X(C)g(1C) = \gamma_g(C). \tag{B.40}$$

Now we prove the error-correction property. Define $\widehat{e}_1(C) \triangleq \widehat{e}_X(C)$, $\widehat{e}_0(C) \triangleq 1 - \widehat{e}_X(C)$, $\widehat{g}_1(C) \triangleq \widehat{g}(1C)$, and $\widehat{g}_0(C) \triangleq \widehat{g}(0C)$. Also, $e_1(C) \triangleq e_X(C)$, $e_0(C) \triangleq 1 - e_X(C)$, $g_1(C) \triangleq g(1C)$, and $g_0(C) \triangleq g(0C)$.

Also, $e_1 = 1 - e_0$ and $\widehat{e}_1 = 1 - \widehat{e}_0$. Then,

$$\mathbb{E}[\zeta_{\widehat{e}_X,\widehat{g}}(XC) \mid C] - \gamma_g(C)$$
$$= \underbrace{\widehat{e}_1\widehat{g}_1 + \widehat{e}_0\widehat{g}_0}_{=A} + \underbrace{\{e_1 - \widehat{e}_1\}\{\widehat{g}_1 - \widehat{g}_0\}}_{=B} - \underbrace{e_1g_1}_{=C} - \underbrace{e_0g_0}_{=D}$$
$$= \underbrace{e_1\{\widehat{g}_1 - g_1\} - e_1\widehat{g}_1}_{=-C} + \underbrace{e_0\{\widehat{g}_0 - g_0\} - e_0\widehat{g}_0}_{=-D} + \underbrace{\widehat{e}_1\widehat{g}_1 + \widehat{e}_0\widehat{g}_0}_{=A} + \underbrace{\{e_1 - \widehat{e}_1\}\{\widehat{g}_1 - \widehat{g}_0\}}_{=B}$$
$$= \widehat{g}_1\{\widehat{e}_1 - e_1\} + \widehat{g}_0\{\widehat{e}_0 - e_0\} + e_1\{\widehat{g}_1 - g_1\} + e_0\{\widehat{g}_0 - g_0\} + \{e_1 - \widehat{e}_1\}\{\widehat{g}_1 - \widehat{g}_0\}.$$

Here,

$$\begin{aligned}
\widehat{g}_1\{\widehat{e}_1 - e_1\} &+ \widehat{g}_0\{\widehat{e}_0 - e_0\} \\
&= \widehat{g}_1\{\widehat{e}_1 - e_1\} - \widehat{g}_0\{\widehat{e}_1 - e_1\} \\
&= \{\widehat{e}_1 - e_1\}\{\widehat{g}_1 - \widehat{g}_0\}.
\end{aligned}$$

Therefore,

$$\mathbb{E}[\varsigma_{\widehat{e}_X,\widehat{g}}(XC) \mid C] - \gamma_g(C) = e_1\{\widehat{g}_1 - g_1\} + e_0\{\widehat{g}_0 - g_0\}. \tag{B.41}$$

## B.7 PROOF OF THEOREM. 3

We first note that

$$\begin{aligned}
\|\widehat{\tau}_{\mathrm{R}} - \tau\|_2^2 &= \|\widehat{b}\widehat{\gamma} - b\gamma_g\|_2^2 \\
&\leq \|\widehat{b}\{\widehat{\gamma} - \gamma_g\} + \gamma_g\{\widehat{b} - b\}\|_2^2 \\
&\lesssim \|\widehat{\gamma} - \gamma_g\|_2^2 + \|\widehat{b} - b\|_2^2.
\end{aligned}$$

By Thm. A.1,

$$\|\widehat{\gamma} - \gamma_g\|_2^2 \leq \mathcal{R}_\Gamma(\epsilon, \widehat{\gamma}, \widehat{\eta}_z) + \{\nabla_\gamma L_\lambda^\gamma(\gamma, \eta_z) - \nabla_\gamma L_\lambda^\gamma(\gamma, \widehat{\eta}_z)\},$$

where, for $h_\gamma \triangleq \|\widehat{\gamma} - \gamma\|_2^2$,

$$\nabla_\gamma L_\lambda^\gamma(\widetilde{\gamma}, \widetilde{\eta}_z)[h_\gamma] = -2\mathbb{E}[\{\varsigma_{\widetilde{\eta}_z}(XC) - \widetilde{\gamma}(C)\}h_\gamma(C)].$$

This gives that $\nabla_\gamma L_\lambda^\gamma(\gamma, \eta_z)[h_\gamma] = 0$. Therefore,

$$\|\widehat{\gamma} - \gamma_g\|_2^2 \leq \mathcal{R}_\Gamma(\epsilon, \widehat{\gamma}, \widehat{\eta}_z) - \nabla_\gamma L_\lambda^\gamma(\gamma, \widehat{\eta}_z),$$

where

$$\begin{aligned}
&- \nabla_\gamma L_\lambda^\gamma(\gamma, \widehat{\eta}_z)[h_\gamma] \\
&= 2\mathbb{E}[\{\varsigma_{\widehat{\eta}_z}(XC) - \widehat{\gamma}(C)\}h_\gamma(C)] \\
&= 2\mathbb{E}[h_\gamma(C)e_X(C)\{\widehat{g}(1C) - g(1C)\} + (1 - e_X(C))\{\widehat{g}(0C) - g(0C)\}] \\
&= 2\|e_1(C)\|_\infty^2 \|h_\gamma(C)\|_2^2 \|\widehat{g}(1C) - g(1C)\|_2^2 + 2\|e_0(C)\|_\infty^2 \|h_\gamma(C)\|_2^2 \|\widehat{g}(0C) - g(0C)\|_2^2 \\
&\leq \alpha\|h_\gamma(C)\|^2 \underbrace{(\|e_1(C)\|_\infty^2 + \|e_0(C)\|_\infty^2)}_{\beta} + \frac{1}{\alpha}\|\widehat{g}(1C) - g(1C)\|_2^2 + \frac{1}{\alpha}\|\widehat{g}(0C) - g(0C)\|_2^2 \\
&\leq \alpha\beta\|h_\gamma(C)\|^2 + \frac{1}{\alpha}\left(\|\widehat{g}(1C) - g(1C)\|_2^2 + \|\widehat{g}(0C) - g(0C)\|_2^2\right).
\end{aligned}$$

Set $\alpha \triangleq \frac{1}{2\beta}$. Then,

$$\begin{aligned}
&- \nabla_\gamma L_\lambda^\gamma(\gamma, \widehat{\eta}_z)[h_\gamma] \\
&\leq \frac{1}{2}\|h_\gamma(C)\|^2 + 2\beta\left(\|\widehat{g}(1C) - g(1C)\|_2^2 + \|\widehat{g}(0C) - g(0C)\|_2^2\right),
\end{aligned}$$

which implies that

$$\frac{1}{2}\|\widehat{\gamma} - \gamma_g\|_2^2 \leq \mathcal{R}_\Gamma(\epsilon, \widehat{\gamma}, \widehat{\eta}_z) + 2\beta\left(\|\widehat{g}(1C) - g(1C)\|_2^2 + \|\widehat{g}(0C) - g(0C)\|_2^2\right),$$

or

$$\|\widehat{\gamma} - \gamma_g\|_2^2 \lesssim \mathcal{R}_\Gamma(\epsilon, \widehat{\gamma}, \widehat{\eta}_z) + \|\widehat{g}(1C) - g(1C)\|_2^2 + \|\widehat{g}(0C) - g(0C)\|_2^2.$$

By Error Analysis of BD-R-Learner,

$$\|\widehat{g}(XC) - g(XC)\|_2^2 \lesssim \mathcal{R}_\mathcal{Q}(\epsilon, \widetilde{g}, \widetilde{\eta}_g) + \|\widehat{e}_Z - e_Z\|_4^4 + \|\widehat{m}_Y - m_Y\|_4^2\|\widehat{e}_Z - e_Z\|_4^2.$$

We note that

$$\|\widehat{g}(XC) - g(XC)\|_2^2 \tag{B.42}$$

$$= \mathbb{E}[\{\widehat{g}(XC) - g(XC)\}^2] \tag{B.43}$$

$$= \sum_{x \in \{0,1\}} \mathbb{E}[\{\widehat{g}(xC) - g(xC)\}^2 \mathbb{I}(X = x)], \tag{B.44}$$

which implies that

$$\mathbb{E}[\{\widehat{g}(xC) - g(xC)\}^2]P(X = x) \leq \mathcal{R}_{\mathcal{Q}}(\epsilon, \widetilde{g}, \widetilde{\eta}_g) + \|\widehat{e}_Z - e_Z\|_4^4 + \|\widehat{m}_Y - m_Y\|_4^2\|\widehat{e}_Z - e_Z\|_4^2,$$

which implies that, for $x \in \{0,1\}$,

$$\|\widehat{g}(xC) - g(xC)\|_2^2 \lesssim \mathcal{R}_{\mathcal{Q}}(\epsilon, \widetilde{g}, \widetilde{\eta}_g) + \|\widehat{e}_Z - e_Z\|_4^4 + \|\widehat{m}_Y - m_Y\|_4^2\|\widehat{e}_Z - e_Z\|_4^2.$$

Therefore,

$$\|\widehat{\gamma} - \gamma_g\|_2^2 \lesssim \mathcal{R}_{\Gamma}(\epsilon, \widehat{\gamma}, \widehat{\eta}_z) + \|\widehat{g}(1C) - g(1C)\|_2^2 + \|\widehat{g}(0C) - g(0C)\|_2^2$$
$$\lesssim \mathcal{R}_{\Gamma}(\epsilon, \widehat{\gamma}, \widehat{\eta}_z) + \mathcal{R}_{\mathcal{Q}}(\epsilon, \widetilde{g}, \widetilde{\eta}_g) + \|\widehat{e}_Z - e_Z\|_4^4 + \|\widehat{m}_Y - m_Y\|_4^2\|\widehat{e}_Z - e_Z\|_4^2.$$

Finally, by Error Analysis of BD-R-Learner,

$$\|\widehat{b}(C) - b(C)\|_2^2 \lesssim \mathcal{R}_{\mathcal{B}}(\epsilon, \widehat{b}, \widehat{\eta}_b) + \|\widehat{e}_X - e_X\|_4^4 + \|\widehat{m}_Z - m_Z\|_4^2\|\widehat{e}_X - e_X\|_4^2.$$

Combining,

$$\|\widehat{\tau}_R - \tau\|_2^2$$
$$\lesssim \|\widehat{\gamma} - \gamma_g\|_2^2 + \|\widehat{b} - b\|_2^2$$
$$= \mathcal{R}_{\Gamma}(\epsilon, \widehat{\gamma}, \widehat{\eta}_z) + \mathcal{R}_{\mathcal{Q}}(\epsilon, \widetilde{g}, \widetilde{\eta}_g) + \mathcal{R}_{\mathcal{B}}(\epsilon, \widehat{b}, \widehat{\eta}_b)$$
$$+ \|\widehat{e}_Z - e_Z\|_4^4 + \|\widehat{m}_Y - m_Y\|_4^2\|\widehat{e}_Z - e_Z\|_4^2 + \|\widehat{e}_X - e_X\|_4^4 + \|\widehat{m}_Z - m_Z\|_4^2\|\widehat{e}_X - e_X\|_4^2,$$

which completes the proof.

# C SIMULATION DETAILS

The full information on implementation is available at https://github.com/yonghanjung/FD-CATE.

## C.1 SYNTHETIC SIMULATION

**Data-generating process (DGP).** For a given sample size $n$ and dimension $d$,

$$C \sim \mathcal{N}(0, I_d), \qquad U \sim \mathcal{N}(0, 1),$$
$$\Pr(X = 1 \mid C, U) = \sigma\big(\beta_0 + \beta_c^\top C + \beta_u U\big),$$
$$\Pr(Z = 1 \mid X, C) = \sigma\big(\alpha_0 + \alpha_c^\top C + \alpha_x X\big),$$
$$Y = \theta_0 + \theta_c^\top C + \theta_z Z + \theta_u U + \varepsilon, \qquad \varepsilon \sim \mathcal{N}(0, 1).$$

We choose "moderate-positivity" coefficients to avoid extreme propensities:

$$\beta_0 = 0.1, \quad \beta_c = 0.7\, w_X, \quad \beta_u = 0.7,$$
$$\alpha_0 = 0.1, \quad \alpha_c = 0.7\, w_Z, \quad \alpha_x = 1.2,$$
$$\theta_0 = 0, \quad \theta_c = 0.7\, w_Y, \quad \theta_z = 1.4, \quad \theta_u = -2.4,$$

where $w_X, w_Z, w_Y \sim \mathcal{N}(0, I_d)$ are $\ell_2$-normalized. The ground-truth heterogeneous effect is

$$\tau_{\text{true}}(c) = \theta_z \Big\{ \sigma\big(\alpha_0 + \alpha_c^\top c + \alpha_x\big) - \sigma\big(\alpha_0 + \alpha_c^\top c\big) \Big\}.$$

**Learning algorithms and cross-fitting.** All nuisances use XGBoost (Chen & Guestrin, 2016) with the same configuration: n_estimators $= 50$, max_depth $= 3$, learning rate $= 0.1$, subsample $= 0.9$, colsample_bytree $= 0.9$, $\ell_2$ penalty $\lambda = 1.0$, histogram tree method. We employ two-fold cross-fitting for FD-PI/FD-DR. FD-R uses a three-way split: nuisances on $\mathcal{D}_1$, $(b, g)$ via BD-R on $\mathcal{D}_2$, and the final $\gamma_g$ regression on $\mathcal{D}_3$, then swap/average. Final regressions on pseudo-outcomes (FD-DR) or on the pseudo-$g$ target (FD-R) use ridge OLS with $\alpha = 10^{-6}$. To control variance, *only denominators* appearing in inverse weights/density ratios are floored at 0.05; numerators are never clipped.

**Structural nuisance noise.** To stress robustness, we inject *rate-level* misspecification at the $n^{-1/4}$ scale:

$$p \in \{e, q\}: \quad p \mapsto \text{clip}_{(0,1)}\big(p + \delta\, \varepsilon\big),$$
$$\mu \in \{m_Y, m\}: \quad \mu \mapsto \mu + \delta\, \varepsilon,$$
$$\varepsilon \sim \mathcal{N}\big(n^{-1/4}, n^{-1/4}\big),$$

sweeping the knob $\delta \in \{0, 0.2, 0.4, 0.6, 0.8, 1.0\}$. Denominators used in weights are *frozen before noise* and floored.

**Weak-overlap stress test.** We steepen the treatment logit via

$$\Pr(X = 1 \mid C, U) = \sigma\big\{\beta_0 + \kappa_e(\beta_c^\top C + \beta_u U)\big\}, \qquad \kappa_e \in \{2, 4, 6, 8, 10\},$$

keeping the mediator regression moderate. This inflates inverse-weight variance (affecting FD-PI/FD-DR) while FD-R remains variance-friendly (no density ratios).

**Design grid and reporting.** Unless noted, $d = 10$. We vary $n \in \{1{,}000, 2{,}500, 5{,}000, 10{,}000, 20{,}000, 50{,}000\}$ and use $R = 100$ Monte Carlo replications. We report RMSE

$$\text{RMSE} = \Big(\mathbb{E}\big[(\widehat{\tau}(C) - \tau_{\text{true}}(C))^2\big]\Big)^{1/2}$$

with mean $\pm$ 95% normal CIs across replications.

## C.2 Real-world study: State seat-belt laws and fatalities (FARS)

**Setting and estimand.** We study the effect of adopting a *primary seat-belt law* on motor-vehicle occupant fatality rates using a state–year panel constructed from the National Highway Traffic Safety Administration's Fatality Analysis Reporting System (FARS) and companion NHTSA survey tables (National Highway Traffic Safety Administration, 2000). Let $C$ denote observed state–year covariates, $X \in \{0,1\}$ indicate whether a primary law is in force, $Z \in \{0,1\}$ be the observed seat-belt use, and $Y$ the *occupant fatality*. Our target is the conditional front–door effect $\tau(C)$.

The FD assumptions are plausible here because (i) the causal pathway from $X$ to $Y$ operates via increased belt use; and (ii) rich $C$ may be sufficient to explain spurious paths between $X$ and $Z$; and $Z$ and $Y$; and (iii) *positivity* holds empirically since belt use is neither zero nor universal in either law regime.

Our result is described in Fig. 3.

**Data and preprocessing.** Following the analysis script (`analyze_fars_2000_fd.py`) provided in supplements, we build a balanced state–year panel and construct $(C, X, Z, Y)$ as follows.

- **Treatment $X$:** indicator that a *primary* seat-belt law is active in a given state and year.
- **Mediator $Z$:** *belt-use* from NHTSA surveys ($\{0,1\}$).
- **Outcome $Y$:** *occupant fatality.*
- **Covariates $C$:** state and year fixed effects and policy-relevant controls compiled at the state–year level: coarse weather severity, and road-class mix and drivers' status.

**Estimators and learning protocol.** We fit the plug-in FD baseline (FD-PI), **FD-DR-Learner**, and **FD-R-Learner** exactly as in the synthetic study: all nuisances are learned with XGBoost (XGBoost; 50 trees, depth 3, learning rate 0.1, subsample/colsample 0.9, $\ell_2$ penalty $\lambda{=}1$); FD-PI/FD-DR use two-fold cross-fitting, and FD-R uses a three-way split (nuisances $\rightarrow (b, g)$ via BD-R $\rightarrow \gamma_g$ via pseudo-$g$), followed by swap-averaging. The final regression on pseudo-outcomes (FD-DR) or on pseudo-$g$ targets (FD-R) uses ridge OLS. To stabilize finite-sample variance we *floor only denominators* that appear in inverse weights/density ratios at 0.05; numerators are never clipped.

