# OpenReview forum: "Debiased Front-Door Learners for Heterogeneous Effects"
_ICLR.cc/2026/Conference — ICLR 2026 Poster_

### Official Review · Reviewer_mEQU · 2025-10-27

**Soundness:** 3
**Presentation:** 2
**Contribution:** 3
**Rating:** 4
**Confidence:** 3

**Summary:**

The authors propose an adaptation of the R-learner for CATE estimation using a front-door structure with a binary mediator (instead of a back-door adjustment which the R-learner was developped for). They fill the gap with prior work that had been focusing on targeted averages, while their method targets conditional effects $\tau(.)$. To do this, they adapt two learners (DR and R-learners) to the FD setting. They reach quasi-oracle

**Strengths:**

The theoretical properties are clear and the setting too. The subject is interesting and well positioned. The empirical evaluation design seems sound.

I especially like the apparent soundness of the results

**Weaknesses:**

I found the paper's notation quite extensive and hard to read. Especially equations 1, 2 and 3 need to be described, and more importantly there is a lack of explanation in the. main text of the nuisance functions defined in Eq. 15, 16 and 17. The authors need to explicitly state what $\bar x$ refers to too.

Why did you not consider comparing your performances with Chen et al. 2025?

I am not sure whether the environments for definitions, propositions and theorems are in accordance to the conference's template

**Questions:**

Could you consider a simulation with controlled FD violation to show the behavior of the estiamtor under misspecification?

Can you give more details on density ratios?

What happens in small sample sizes? The sizes considered in the synthetic study are very big and FD-PI seems to have better RMSE in lower sample sizes, can you comment on this?

Can your method provide confidence intervals for $\tau(.)$?

Other baselines could be considered in the experiments such as the oracle estimator and the mentionned Chen et al. 2025 method.

---

> ### Author Response · Authors · 2025-11-20
> **Response to Reviewer mEQU**
>
> We thank Reviewer mEQU for their insightful comments.
>
> ---
>
> > W1: I found the paper's notation quite extensive and hard to read. Especially equations 1, 2 and 3 need to be described, and more importantly there is a lack of explanation in the. main text of the nuisance functions defined in Eq. 15, 16 and 17. The authors need to explicitly state what  refers to too.
> >
>
> We thank the reviewer for this constructive feedback on presentation. In the revised manuscript, we will expand the text around all defining equations (1-3 and 15-17) to ensure they are clearly defined for the reader:
>
> - **$m(zxc)$ (Eq. 1):** Outcome model, representing the expected outcome $Y$ given the mediator $Z$, treatment $X$, and covariates $C$.
> - **$e(x|c)$ (Eq. 2):** Treatment propensity score, or the probability of receiving treatment $X$ given covariates $C$.
> - **$q(z|xc)$ (Eq. 3):** Mediator propensity score, or the probability of the mediator $Z$ taking its value given treatment $X$ and covariates $C$.
> - **$r_{me}$ (Eq. 15):** Expected outcome conditional on the mediator ($Z$) and covariates ($C$), after marginalizing out the treatment ($X$).
> - **$\nu_{meq}$ (Eq. 16):** Expected value of $r_{me}$ after *further* marginalizing out the mediator ($Z$).
> - **$s_{mq_{\overline{x}}}$ (Eq. 17):** Expected outcome under a *counterfactual* distribution of the mediator (i.e., $q(z|\overline{x}C)$).
>
> ---
>
> > W2: Why did you not consider comparing your performances with Chen et al. 2025?
> >
>
> > Q5. Other baselines could be considered in the experiments such as the oracle estimator and the mentionned Chen et al. 2025 method.
> >
>
> We have conducted new experiments comparing our methods to this work and discuss the result in **Common Response 1: Comparison with Chen et al. (LobsterNet)**.
>
> ---
>
> > W3: I am not sure whether the environments for definitions, propositions and theorems are in accordance to the conference's template
> >
>
> We have double-checked the ICLR 2026 author guidelines and confirm that our theorem and definition environments is aligned with the conference's template.
>
> ---
>
> > Q1: Could you consider a simulation with controlled FD violation to show the behavior of the estiamtor under misspecification?
> >
>
> While Figure 2d already illustrates robustness against near-violations of the overlap assumption (positivity), we have conducted an additional simulation to test robustness against unmeasured confounding violation.
>
> Specifically, we modify the data generating process such that the mediator is affected by the hidden confounder through a coefficient (Conf. Strength) ranging from [0.0, 0.15, 0.3, 0.45, 0.6]. The results show the RMSE of the estimated effect:
>
> | **Confound Strength** | **FD-PI** | **FD-DR** | **FD-R** |
> | --- | --- | --- | --- |
> | 0.00 | 0.016 ± 0.006 | 0.007 ± 0.001 | 0.008 ± 0.002 |
> | 0.15 | 0.027 ± 0.003 | 0.012 ± 0.002 | 0.013 ± 0.002 |
> | 0.30 | 0.046 ± 0.003 | 0.024 ± 0.003 | 0.025 ± 0.003 |
> | 0.45 | 0.088 ± 0.002 | 0.063 ± 0.002 | 0.060 ± 0.003 |
> | 0.60 | 0.108 ± 0.004 | 0.086 ± 0.008 | 0.086 ± 0.010 |
>
> As shown, the FD-DR and FD-R learners exhibit lower RMSE and are more robust against the FD assumption violation (unmeasured confounding) compared to the FD-PI estimator.
>
> ---
>
> > Q2. Can you give more details on density ratios?
> >
>
> You are referring to the density ratios $\pi_{\overline{x}}(XC)$ and $\xi_{\overline{x}}(ZXC)$. Currently, we estimate these via a plug-in approach using $\hat{q}$ and $\hat{e}$.
>
> To control variance, users may employ truncated density ratios. Alternative estimation strategies could be used. For the ratio $\xi_{\overline{x}} = q(Z|\overline{x}C) / q(Z|XC) = \frac{P(\overline{x} \mid Z, C) P(X \mid C)}{P(X \mid Z, C) P(\overline{x} \mid C)}$
>
> If estimating the components on the right-hand side is easier than directly estimating $q(Z|XC)$, this alternative decomposition could lead to more stable estimates. Developing more robust density ratio techniques remains an important future direction.
>
> ---
>
> > Q3. What happens in small sample sizes? The sizes considered in the synthetic study are very big and FD-PI seems to have better RMSE in lower sample sizes, can you comment on this?
> >
>
> We have conducted new simulation experiments focusing on the small sample regime and reported in **Common Response 2: Comparison with Small Sample**.
>
> ---
>
> > Q4. Can your method provide confidence intervals for ?
> >
>
> The error bars in Figure 2 show the 2.5%–97.5% empirical quantiles of the proposed estimators across Monte Carlo replications; we will clarify in the paper that these are Monte Carlo bands.
>
> We note that this is not the confidence interval (CI), and for CATE estimators, deriving the CI is non-trivially difficult. Specifically, our estimator converges at best at the oracle rate, not $o_P(n^{-1/2})$. As a result, usual root-n asymptotic normality is not expected unless we make strong assumptions. Deriving the CI is out of scope of this work, and we view this as an important future direction.

---

### Official Review · Reviewer_KeNg · 2025-10-27

**Soundness:** 3
**Presentation:** 2
**Contribution:** 3
**Rating:** 6
**Confidence:** 3

**Summary:**

This paper considers the front-door criterion to identify heterogeneous treatment effect where there exists unobserved confounding. The authors proposed two estimator for this regime: FD-DR-learner and FD-R-learner. The FD-DR-learner follows the DR-learner framework that constructs a doubly robust pseudo-outcome leveraging FD identification of the treatment effect first, then regresses the covariates on the pseudo-outcome. The FD-R-learner first uses the (backdoor) R-learner to fit some of the nuisances, then constructs a "pseudo-outcome/function" from the learned nuisances to obtain the final estimator. Moreover, the author also provides an error analysis for the proposed estimator.

**Strengths:**

- Proposed two novel estimators with discussion on their performance, and when one is preferred over the other.

- Provided error analysis that demonstrated the robustness of the proposed estimators.

**Weaknesses:**

- Proposed estimators are mainly built on top of ideas from exiting estimators.

- The FD-R-learner requires the estimation of many nuisance functions (has to split data into 3 folds). This could suffer when the sample size is small.

- The paper discussed cases where one estimator is preferred over the other, but did not include experiments to support the claims.

**Questions:**

- This paper only considers binary mediators, can the estimators be extended to higher dimensional or continuous mediators?

---

> ### Author Response · Authors · 2025-11-20
> **Response to Reviewer KeNg**
>
> We thank Reviewer KeNg for their thorough assessment and insightful comments. We address each point below.
>
> ---
>
> > W1: Proposed estimators are mainly built on top of ideas from existing estimators.
>
> We thank the reviewer for this observation. The perceived "similarity" is a consequence of the FD structure itself, which can be seen as a composition of two BD problems. Our work formalizes this complexity:
>
> 1. **FD-DR-Learner:** While the spirit of regressing a doubly robust pseudo-outcome is the same, the derivation and analysis are significantly more complex. The FD pseudo-outcome (Definition 2) and its double robustness property (Lemma 2) depend on three distinct nuisance functions ($m, e, q$), unlike the two found in the standard BD setting. Proving that this estimator still achieves quasi-oracle rates (Theorem 1) is a non-trivial theoretical contribution.
> 2. **FD-R-Learner**: This estimator is substantially more novel8. Its derivation required us to first introduce a new partial linear model representation for the FD data-generating process (Proposition 2), which had not been previously studied9. We then show how this new structure can be solved by a novel, multi-stage procedure (Definition 5) that strategically uses BD-R-Learners as debiased components to estimate the pathway effects ($b(C)$ and $g(XC)$).
>
> Thus, while we leverage existing ideas, the core of our paper lies in the new theoretical results (Proposition 2, Theorem 1, Theorem 3) required to make these estimators work in the front-door setting.
>
> ---
>
> > W2: The FD-R-learner requires the estimation of many nuisance functions (has to split data into 3 folds). This could suffer when the sample size is small.
>
> We thank the reviewer for pointing out the vulnerability of the multi-stage FD-R learner to small sample sizes. In response to this and a similar comment from Reviewer mEQU, we have conducted new simulation experiments focusing on the small sample regime and reported in **Common Response 2: Comparison with Small Sample**.
>
> ---
>
> > W3: The paper discussed cases where one estimator is preferred over the other, but did not include experiments to support the claims.
>
> We believe our paper does provide direct experimental support for our claims in Section 4.1 regarding when to prefer one estimator over the other. Our primary guidance is to prefer the FD-R-Learner when overlap is weak. This claim is tested and directly validated by our experiment in Figure 2d ("Weak-overlap severity"). Figure 2d shows that as overlap violation increases, the FD-DR's variance inflates severely due to inverse weights, while the FD-R-Learner remains stable and outperforms both baselines.
>
> ---
>
> > Q1: This paper only considers binary mediators, can the estimators be extended to higher dimensional or continuous mediators?
>
> This is an excellent question and a critical direction for future research. Our current theory focuses on the binary-mediator case because it is the most fundamental but unexplored setting in estimating heterogeneous treatment effects in FD models.
>
> Extending our debiased learners to continuous or multidimensional mediators is indeed a highly nontrivial challenge. It would require replacing the discrete summations in our core definitions (e.g., in the FD-DR pseudo-outcome and the FD-R partial linear model ) with complex integral equations, which introduces significant new theoretical and computational hurdles. We view our work as a foundational step toward this broader goal, and we plan to pursue these generalizations in future research.

---

### Official Review · Reviewer_sZgx · 2025-10-30

**Soundness:** 3
**Presentation:** 3
**Contribution:** 2
**Rating:** 4
**Confidence:** 3

**Summary:**

This paper tackles the challenge of estimating heterogeneous treatment effects (HTEs) under unmeasured confounding using the front-door (FD) criterion. It introduces two novel debiased estimators—FD-DR-Learner and FD-R-Learner—that achieve quasi-oracle convergence rates even when nuisance functions converge slowly at $n^{-1/4}$. FD-DR leverages doubly robust pseudo-outcomes, while FD-R decomposes effects into interpretable causal pathways. Both methods demonstrate superior performance and robustness in synthetic and real-world experiments, providing accurate and stable HTE estimates where traditional plug-in estimators fail. Overall, the study advances causal inference by enabling reliable, personalized effect estimation under front-door settings with theoretical and empirical validation.

**Strengths:**

1. This paper is among the first to tackle HTEs under FD.
2. The proposed method is supported by rigorous theoretical analysis.
3. Experimental results demonstrate that the proposed approach.

**Weaknesses:**

1. The proposed methods and theory are restricted to binary mediators.
2. The paper does not include empirical comparisons with existing HTE-FD methods (e.g., Chen et al., 2025, LobsterNet).
3. Beyond double robustness, a key advantage of debiased estimators lies in their asymptotic normality, which enables valid statistical inference. However, the paper does not provide any inference results or variance estimation, leaving the uncertainty quantification of the proposed estimators unexplored.

**Questions:**

1. While theoretically appealing, the front-door identification relies on strong and often unverifiable assumptions—such as the absence of unmeasured confounders between mediators and outcomes. As noted by Imbens and others [1], these conditions are rarely satisfied in practice, raising concerns about the empirical credibility and real-world applicability of the proposed methods.
2.  Figure 3(a)’s histogram suggests that the sampling distribution of the estimator deviates noticeably from normality.



[1] Potential Outcome and Directed Acyclic Graph Approaches to Causality: Relevance for Empirical Practice in Economics

---

> ### Author Response · Authors · 2025-11-20
> **Response to Reviewer sZgx**
>
> We thank Reviewer sZgx for constructive feedback and insightful questions.
>
> ---
>
> > W1: The proposed methods and theory are restricted to binary mediators.
> >
>
> Thank you for pointing out important limitation. Our current theory focuses on the binary-mediator case because it is the most fundamental but unexplored setting in estimating heterogeneous treatment effects in FD models. Extending our framework to continuous or high-dimensional mediators is indeed a valuable and nontrivial direction. We view our work as a foundational step toward this broader goal, and we plan to pursue these generalizations in future research.
>
> ---
>
> > W2: The paper does not include empirical comparisons with existing HTE-FD methods (e.g., Chen et al., 2025, LobsterNet).
> >
>
> We thank the reviewer for this suggestion. In response, we have conducted new experiments comparing our methods to this work. We respectfully refer the reviewer to the **Common Response 1: Comparison with Chen et al. (LobsterNet)** for a detailed discussion and empirical comparison. This comparison will be added in the appendix of the revised manuscript.
>
> ---
>
> > W3: Beyond double robustness, a key advantage of debiased estimators lies in their asymptotic normality, which enables valid statistical inference. However, the paper does not provide any inference results or variance estimation, leaving the uncertainty quantification of the proposed estimators unexplored.
> >
>
> This is an excellent point. We agree that uncertainty quantification (UQ) is a critical aspect of estimation. In Fig. 2, the error bars represent the 2.5%–97.5% *empirical quantiles* across Monte Carlo replications (*Monte Carlo variability)* to empirically capture the uncertainty.
>
> Deriving valid, practical confidence intervals (CIs) for non-parametric CATE estimators is a highly non-trivial research problem. As our estimators converge at a quasi-oracle rate (which is typically slower than $n^{-1/2}$), standard root-n asymptotic normality is not expected. This makes conventional CI derivation challenging. Developing valid UQ for these debiased learners is a significant challenge in its own right and, while out of scope for the present work, a key direction for future research.
>
> ---
>
> > Q2: Figure 3(a)’s histogram suggests that the sampling distribution of the estimator deviates noticeably from normality.
> >
>
> The histogram in Figure 3(a) 9 does not represent a *sampling distribution* of a single estimate. Rather, it shows the distribution of the estimated heterogeneous effects ($\hat{\tau}(C_i)$) across all individuals in the FARS dataset. So, this distribution is not necessarily follow the normal distribution.
>
> ---
>
> > Q1: While theoretically appealing, the front-door identification relies on strong and often unverifiable assumptions—such as the absence of unmeasured confounders between mediators and outcomes. As noted by Imbens and others [1], these conditions are rarely satisfied in practice, raising concerns about the empirical credibility and real-world applicability of the proposed methods.
> >
>
> We appreciate this fundamental question about the FD criterion's practical utility. The FD models are not merely theoretical, because they are actively used in practice:
>
> 1. Piccininni et al. (2023)  [1] used the FD formula to estimate the causal effect of mobile stroke units in a high-impact epidemiological study, recognized its practicality with a **Rothman Epidemiology Prize** in the field of epidemiology.
> 2. Inoue et al. (2022) [2] applied the FD formula to study the causal effect of chronic pain on mortality through opioid prescriptions using real-world public health dataset.
> 3. Glynn & Kashin (2018) [3] used the real-world data analysis to demonstrate that FD approach can yield substantially *less bias* than back-door methods when confounding between treatment and outcome is strong, and mediator–outcome confounding is weaker.
>
> In summary, the FD criterion is an important and increasingly adopted tool in fields like epidemiology because of its plausibility in capturing reality under specific confounding structures.
>
> ---
>
> [1] Piccininni, M., Kurth, T., Audebert, H. J., and Rohmann, J. L. (2023). The effect of mobile stroke unit care on functional outcomes: an application of the front-door formula. Epidemiology, 34(5):712–720.
>
> [2] Inoue, Kosuke, Beate Ritz, and Onyebuchi A. Arah. "Causal effect of chronic pain on mortality through opioid prescriptions: application of the front-door formula." *Epidemiology* 33.4 (2022): 572-580.
>
> [2] Glynn, Adam N., and Konstantin Kashin. "Front-door versus back-door adjustment with unmeasured confounding: Bias formulas for front-door and hybrid adjustments with application to a job training program." *Journal of the American Statistical Association* 113.523 (2018): 1040-1049.

---

### Official Review · Reviewer_YM1c · 2025-10-31

**Soundness:** 4
**Presentation:** 4
**Contribution:** 3
**Rating:** 8
**Confidence:** 4

**Summary:**

This work presents estimators for the CATE under a front-door adjustment formula. This contrasts with existing works, which focus on a back-door formula. The authors derive variants of the DR- and R-leaners for this setting and provide an error analysis that reveals double robustness.

**Strengths:**

To my knowledge, this the first estimator of the CATE under front-door adjustment. This is important since it makes it possible to estimate the CATE in a wider variety of settings than was previously possible.

The theoretical arguments appear to be sound.

**Weaknesses:**

The theoretical analysis ends up being somewhat similar to that of existing backdoor methods, but that's no fault of the authors - that analysis strategy works well.

**Questions:**

Minor comment on references on line 41-42:

* R-learner was first proposed in Corollary~9.1 of
Robins, James M. "Optimal structural nested models for optimal sequential decisions." Proceedings of the Second Seattle Symposium in Biostatistics: analysis of correlated data. New York, NY: Springer New York, 2004.
* DR-learner was proposed in Section 3.1 of
van der Laan, Mark J. "Targeted Learning of an Optimal Dynamic Treatment, and Statistical Inference for its Mean Outcome." (2013).

---

> ### Author Response · Authors · 2025-11-20
> **Response to Reviewer Ym1c**
>
> We are extremely grateful to the reviewer for their positive assessment and high rating. We sincerely appreciate that you recognized our work as the first to develop CATE estimators for the front-door setting and found our theoretical arguments to be sound.
>
> We also thank the reviewer for their very helpful suggestions regarding the references for the R-learner and DR-learner. In our revised manuscript, we update our related work section to include these foundational citations (Robins, 2004 and van der Laan, 2013) to give proper credit to their original development.
>
> Regarding the theoretical analysis, we thank the reviewer for this observation. We agree the analysis strategy is inspired by existing back-door (BD) methods, which is a natural fit as the front-door (FD) structure can be viewed as a composition of two BD adjustments. We appreciate the reviewer's recognition that this analysis strategy works well for our setting.

---

> > ### Comment · Reviewer_YM1c · 2025-11-25
> >
> > Thank you for your replies. After reading through them and the assessments of the other reviewers, I will leave my score as is.

---

### Author Response · Authors · 2025-11-20
**Common Response 1: Comparison with Chen et al. (LobsterNet)**

Reviewers sZgx and mEQU asked about comparisons with Chen et al. (2025) [1], who proposed LobsterNet, a neural-network–based estimator for heterogeneous front-door effects. A key clarification is that LobsterNet is a *plug-in front-door estimator (FD-PI)*, where the nuisance components are modeled using a neural network. Their method does **not** incorporate debiasing, orthogonality, or quasi-oracle guarantees. Since our proposed estimators are agnostic to any nuisance learners, our Theorem (1,3) guarantees outperformance of FD-DR and FD-R learners. Indeed, you may think our estimators as a principled way of enhancing Neural-network based FD-CATE learners such as LobsterNet.

Given this, we compared our debiased estimators (FD-DR and FD-R) to a plug-in baseline under the same simulation settings as Chen et al. (2025), using sample sizes n=[1000, 2500, 5000, 10000, 20000, 50000]

---

**1. Simulation without nuisance noise**

| **n** | **FD-PI** | **FD-DR** | **FD-R** |
| --- | --- | --- | --- |
| 1000 | 0.228 ± 0.044 | 0.228 ± 0.029 | 0.292 ± 0.025 |
| 2500 | 0.165 ± 0.024 | 0.206 ± 0.035 | 0.188 ± 0.022 |
| 5000 | 0.142 ± 0.026 | 0.143 ± 0.016 | 0.138 ± 0.011 |
| 10000 | 0.108 ± 0.015 | 0.093 ± 0.007 | 0.086 ± 0.005 |
| 20000 | 0.096 ± 0.006 | 0.077 ± 0.004 | 0.075 ± 0.003 |
| 50000 | 0.078 ± 0.010 | 0.055 ± 0.003 | 0.054 ± 0.002 |

As predicted by Theorems 1 and 3, **FD-DR** and **FD-R** outperform the plug-in estimator as n increases.

---

**2. Simulation with nuisances converging at the $n^{-1/4}$ rate.**

| **n** | **FD-PI** | **FD-DR** | **FD-R** |
| --- | --- | --- | --- |
| 1000 | 0.341 ± 0.022 | 0.294 ± 0.046 | 0.263 ± 0.035 |
| 2500 | 0.248 ± 0.022 | 0.183 ± 0.017 | 0.181 ± 0.017 |
| 5000 | 0.203 ± 0.008 | 0.147 ± 0.020 | 0.140 ± 0.016 |
| 10000 | 0.162 ± 0.007 | 0.100 ± 0.009 | 0.097 ± 0.007 |
| 20000 | 0.133 ± 0.006 | 0.079 ± 0.006 | 0.072 ± 0.003 |
| 50000 | 0.104 ± 0.004 | 0.053 ± 0.002 | 0.054 ± 0.002 |

These results demonstrate the robustness of the debiased learners. Even when nuisances converge slowly, **FD-DR** and **FD-R** maintain nearly the same performance as in the noiseless case, which achieves precisely the quasi-oracle behavior guaranteed by their orthogonal structure (Theorems 1,3). The plug-in estimator degrades substantially under this setting, consistent with the theoretical expectation.

---

**Takeaway**

LobsterNet is a FD-PI with neural nuisance learners, without debiasing or orthogonality. Our methods provides principled method to enhance those non-debiased estimators. As gauranteed in Theorems (1,3), our comparison shows that **FD-DR** and **FD-R** consistently improve upon plug-in estimators, owing to their debiased construction and quasi-oracle error control.

---

[1] Winston Chen, Trenton Chang, Jenna Wiens Proceedings of the sixth Conference on Health, Inference, and Learning, PMLR 287:194-230, 2025.

---

### Author Response · Authors · 2025-11-20
**Common Response 2: Comparison with Small Sample**

This response addresses the requests from Reviewers KeNg and mEQU regarding the performance of our estimators in small sample regimes. We conducted new simulations comparing our debiased estimators (FD-DR and FD-R) to the plug-in baseline (FD-PI) using small sample sizes, specifically $n=[100, 250, 500, 1000, 2000, 5000]$.

The results confirm a known trade-off in orthogonal debiased machine learning: FD-PI temporarily outperforms FD-DR and FD-R at very small $n$ due to the high finite-sample variance imposed by cross-fitting. However, our debiased learners quickly dominate once sample size increases, where their theoretically guaranteed bias reduction takes effect.

---

**1. Simulation without nuisance noise**

| **n** | **FD-PI** | **FD-DR** | **FD-R** |
| --- | --- | --- | --- |
| 100 | 0.348 ± 0.018 | 1.092 ± 0.618 | 2.213 ± 0.382 |
| 250 | 0.315 ± 0.020 | 0.644 ± 0.250 | 0.606 ± 0.051 |
| 500 | 0.271 ± 0.015 | 0.300 ± 0.042 | 0.432 ± 0.049 |
| 1000 | 0.212 ± 0.007 | 0.272 ± 0.030 | 0.267 ± 0.022 |
| 2000 | 0.185 ± 0.016 | 0.188 ± 0.028 | 0.202 ± 0.025 |
| 5000 | 0.147 ± 0.021 | 0.125 ± 0.014 | 0.144 ± 0.021 |

**2. Simulation with nuisances converging at the $n^{-1/4}$ rate.**

| **n** | **FD-PI** | **FD-DR** | **FD-R** |
| --- | --- | --- | --- |
| 100 | 0.394 ± 0.023 | 2.057 ± 0.559 | 1.679 ± 0.467 |
| 250 | 0.404 ± 0.020 | 0.901 ± 0.181 | 0.579 ± 0.097 |
| 500 | 0.356 ± 0.039 | 0.446 ± 0.039 | 0.359 ± 0.049 |
| 1000 | 0.307 ± 0.021 | 0.307 ± 0.036 | 0.233 ± 0.043 |
| 2000 | 0.264 ± 0.005 | 0.215 ± 0.033 | 0.201 ± 0.025 |
| 5000 | 0.196 ± 0.007 | 0.125 ± 0.017 | 0.137 ± 0.013 |

---

For $n \le 500$, the FD-PI baseline yields lower error and lower variance than our debiased learners. This degradation is expected and can be explained by the necessary structural overhead of orthogonal methods:

1. **Cross-Fitting Penalty:** Both FD-DR (2-split) and especially FD-R (3-split) require splitting the data to ensure orthogonality and manage asymptotic bias. At $n=100$, the training samples used for the nuisance functions (e.g., XGBoost) are severely reduced ($n/2$ or $n/3$), leading to poorly estimated nuisance models and unreliable pseudo-outcomes. The FD-PI estimator temporarily avoids this penalty by using the entire dataset.
2. **Variance Inflation (FD-DR):** FD-DR uses inverse density weights (density ratios $\xi_{\overline{x}}$). In small samples, these denominators are poorly estimated, causing the weights to inflate drastically, resulting in the variance explosion seen at $n=100$.
3. **Nuisance Error Multiplication:** Theorems 1 and 3 establish that FD-DR and FD-R achieve quasi-oracle rates because their errors are controlled by higher-order products of nuisance errors (e.g., $||\hat{\eta}_1 - \eta_1|| \times ||\hat{\eta}_2 - \eta_2||$ or $||\hat{\eta} - \eta||^4$). This multiplication ensures that if the individual nuisance errors are $\mathcal{O}(n^{-1/4})$, the total bias term is $\mathcal{O}(n^{-1/2})$ or higher, effectively making the debiased estimator converge at the fast oracle rate. However, if the nuisance errors are large (as they are at $n=100$), then the error terms are multiplied into larger, rather than smaller, values, causing initial error inflation. Since flexible machine learning estimators like XGBoost or Neural Networks (NNs) are used, as data grows, we expect these nuisance errors to become moderate, allowing the quasi-oracle properties of FD-DR and FD-R to dominate for $n \ge 1000$.

---

### Meta-Review · Area_Chair_CU1c · 2026-01-07

**Summary:**

This is a difficult case -- most of the reviews are borderline and do not engage with the paper in much depth. Nevertheless I am recommending acceptance since the weaknesses pointed out in the negative reviews are either not significant or have been addressed in the rebuttal.

**Reviewer Concerns:**

Several reviewers wanted to see a comparison to Chen et al., 2025, LobsterNet. The authors have added such a comparison at the rebuttal stage.

Other concerns were more minor (e.g. extensions to non-binary mediators, asymptotic normality, etc.) and mostly left to future study by the authors.

**Reviewer Scores:**

Reviewers sZgx and mEQU may have increased their scores by 1 or so due to the added comparision with LobsterNet, which was a concern both shared.

---

### Decision · Program_Chairs · 2026-01-26

Accept (Poster)